# Successor Heads: Recurring, Interpretable Attention Heads In The Wild

**Rhys Gould**[1], **Euan Ong**[1], **George Ogden**[1], **Arthur Conmy**[2]
[1] University of Cambridge     [2] Independent
Correspondence to `rg664@cam.ac.uk`

## Abstract

In this work we describe successor heads: attention heads that increment tokens with a natural ordering, such as numbers, months, and days. For example, successor heads increment 'Monday' into 'Tuesday'. We explain the successor head behavior with an approach rooted in mechanistic interpretability, the field that aims to explain how models complete tasks in human-understandable terms. Existing research in this area has struggled to find recurring, mechanistically interpretable large language model (LLM) components beyond small toy models. Further, existing results have led to very little insight to explain the internals of the larger models that are used in practice. In this paper, we analyze the behavior of successor heads in LLMs and find that they implement abstract representations that are common to different architectures. Successor heads form in LLMs with as few as 31 million parameters, and at least as many as 12 billion parameters, such as GPT-2, Pythia, and Llama-2. We find a set of 'mod 10' features[1] that underlie how successor heads increment in LLMs across different architectures and sizes. We perform vector arithmetic with these features to edit head behavior and provide insights into numeric representations within LLMs. Additionally, we study the behavior of successor heads on natural language data, where we find that successor heads are important for achieving a low loss on examples involving succession, and also identify interpretable polysemanticity in a Pythia successor head.

## 1 Introduction

Figure 1: A successor head with OV matrix $W_{OV}$ takes a numbered token such as 'Monday' in embedding space and maps it to its successor value in unembedding space, e.g. 'Tuesday'. The circuit is the simple composition of the embedding matrix, the first MLP block, a single attention head, and the unembedding matrix.

Mechanistic interpretability (Olah, 2022) is the process of reverse-engineering the algorithms that trained neural networks have learned. Recently, much attention has been paid to interpreting transformer-based large language models (LLMs), as these models have demonstrated impressive capabilities (OpenAI, 2023) but there is little understanding of how these models produce their outputs. Existing interpretability research includes comprehensive reverse-engineering efforts into toy models (Nanda et al., 2023) and small language models (Olsson et al., 2022; Wang et al., 2023), though few insights have been gained about how frontier LLMs function.

---

[1] In this work, we use 'feature' to mean an interpretable (linear) direction in activation space, inspired by the second 'potential working definition' from Elhage et al. (2022).

In mechanistic interpretability, universality (Olah et al., 2020; Li et al., 2016) is a hypothesis that there are *common representations* in neural networks. The universality hypothesis asserts that neural networks with different architectures and *scales* form common internal representations. Strong evidence for (or against) the universality hypothesis could significantly affect research priorities in interpretability. If common representations form across different language models and tasks, then research on small or toy language models (Elhage et al., 2022; 2021) and narrow tasks (Wang et al., 2023; Heimersheim & Janiak, 2023; Hanna et al., 2023) may be the best way to gain insights into LLM capabilities. Conversely, if the representations used by language models do not generalize to different model scales and/or tasks, then developing methods that can be applied to larger language models and don't rely on lessons from small models generalizing (such as Wu et al. (2023), Bills et al. (2023), Conmy et al. (2023)) may be the most fruitful direction for interpretability.

In this work, we find an interpretable set of attention heads we call **successor heads** in models of many different scales and architectures. Successor heads are attention heads that perform incrementation in language models. The input to a successor head is the representation of a token in an ordinal sequence such as 'Monday', 'first', 'January', or 'one'. The output of a successor head assigns a higher likelihood to the incremented token, such as 'Tuesday', 'second, 'February', or 'two'. In our work, we find evidence for a weak form of universality (Chughtai et al. (2023); points 1. and 2.) in finding Successor Heads across different models, as well as finding that numeric representations in language models are compositional (point 3.), as

1. Successor heads form across language models from the *scale* of 31 million parameters and to at least as many as 12 billion parameters.
2. Successor heads form across models with different architectures, including Pythia, GPT-2 and Llama-2 (Touvron et al., 2023).
3. Language models use *compositional numeric representations* to encode the index of these tokens within their ordinal sequence; these representations exhibit *abstract structure*, such as mod-10 features.

Our contributions can be summarised as follows:

**1. Introducing and interpreting successor heads (Section 2-3)**

(a) We introduce and explain successor heads, and show that they occur in language models across almost three orders of magnitude of model parameter count.

**2. Finding abstract numeric representations in language models (Section 3)**

(b) We isolate a common *numeric subspace* within embedding space, that for any given token (e.g. 'February') encodes the index of that token within its ordinal sequence (e.g. months).

(c) We find evidence for *interpretable, abstract features* within successor heads' numeric inputs: an unsupervised decomposition of token representations yields a crucial set of features we call the **mod-10 features** $\{f_0, ..., f_9\}$. $f_n$ is present in all tokens whose numerical index $\equiv n \pmod{10}$, e.g. $f_2$ is present in the model's representations of '2', '32', '172', 'February', 'second' and 'twelve'.

(d) We steer the semantics of successor heads' numeric inputs with vector arithmetic.

**3. Showing that the succession mechanism is important in the wild (Section 4)**

(d) Finally, we show that successor heads play an important role in incrementation-based tasks in natural language datasets – for instance, predicting the next number in a numbered list of items.

## 2 SUCCESSOR HEADS

LLMs are able to increment elements in an **ordinal sequence**. For instance, Pythia-1.4B will complete the prompt "If this is 1, the next is" with " 2", and the prompt "If this is January, the next is" with " February". Given this observation, we find evidence for attention heads within LLMs (which we refer to as **successor heads**) responsible for performing this type of incrementation. To get evidence for successor heads we require three definitions: i) the **succession dataset** of tasks involving abstract numeric representations, ii) an **effective OV circuit** to measure how attention heads affect model outputs, and finally iii) **successor score**.

The ***succession dataset*** is the set of tokens across eight different *tasks* that can be incremented:[2]

| Task | Tokens |
|------|--------|
| Numbers | '1', '2', ..., '199', '200' |
| Number words | 'one', 'two', ..., 'nineteen', 'twenty' |
| Cardinal words | 'first', 'second', ..., 'tenth' |
| Days | 'Monday', 'Tuesday', ..., 'Sunday' |
| Day prefixes | 'Mon', 'Tue', ..., 'Sun' |
| Months | 'January', 'February', ..., 'December' |
| Month prefixes | 'Jan', 'Feb', ..., 'Dec' |
| Letters | 'A', 'B', ..., 'Z' |

Table 1: Tokens in the succession dataset

We also include different forms of these tokens, as language model tokenizers often have several tokens corresponding to the same word (e.g. words with/without a space at the start being different tokens). Full details of our dataset can be found in our open-sourced experiments.[3]

**Notation**. For consistency with prior work we follow all Elhage et al. (2021)'s notation choices, though the following definitions are sufficient and self-contained for this work regardless. Transformer language models use an embedding matrix $W_E \in \mathbb{R}^{d_{\text{model}} \times n_{\text{vocab}}}$ to map tokens to vectors in the ***residual stream*** (the cumulative sum of embeddings, attention heads and MLPs). After additive application of attention and MLP layers, the unembedding matrix $W_U \in \mathbb{R}^{n_{\text{vocab}} \times d_{\text{model}}}$ maps the final state of the residual stream to logits for all next token predictions. An OV matrix $W_{OV} \in \mathbb{R}^{d_{\text{model}} \times d_{\text{model}}}$ maps the residual stream to the output of an attention head, assuming the attention head solely attended to that residual stream. Altogether, our diagram in Figure 1 shows one shallow path through a transformer model.

**Effective OV Circuit**. We determine whether attention heads perform succession by studying their effective OV circuit, which measures the ***direct effect*** of input tokens when multiplied by an OV matrix $W_{OV}$, as in concurrent work (McDougall et al., 2023) which surveys the importance of MLP0. The (non-effective) OV circuit $W_U W_{OV} W_E$ (1) from Elhage et al. (2021) is the inspiration for our ***effective OV circuit*** $W_U W_{OV} \text{MLP}_0(W_E)$ (2). Intuitively, (2)'s columns represent input tokens to the head and the rows represent the logits on each possible output token.

**Successor Heads** are then operationalized by considering an input token $t$ from our succession dataset (e.g. $t = $ 'Monday'). If the effective OV circuit column for input $t$ has a larger output on the successor to $t$ ('Tuesday') than on any other of the tokens in that task ('Monday' or 'Wednesday' or 'Thursday' or ...) then we consider the head to have performed succession in this case. ***Successor Heads*** are then defined as the attention heads that pass this test for more than half of the tokens in the succession dataset. We call the proportion of succession dataset tokens on which an attention head performs succession the ***succession score***. The succession scores across a range of models are displayed in Figure 2.

## 3 DECOMPOSING NUMERIC REPRESENTATIONS

Having found behavioral evidence that successor heads exist across a range of models, we now perform a case study on the attention head (L12H0) with the maximal successor score in Pythia-1.4B. Indeed, not only do we find that the representations on which the successor head acts (i.e. the outputs of the $\text{MLP}_0$ layer) share a common *numeric subspace* (Section A), that for any given token (e.g. 'February') encodes the index of that token within its ordinal sequence (e.g. months), but we also find mechanistic evidence for *transferable arithmetic features* within these representations.[4]

---

[2]The days and months tasks are special as the final tokens in these classes ('Sunday' and 'December') have cyclical successors ('Monday' and 'January'). We don't consider the end tokens of the other tasks to have cyclical successors.

[3]Available soon at `https://github.com/euanong/numeric-representations/blob/main/exp_numeric_representations/model.py#L19`

[4]Note that we also observe similar abstract numeric representations across other models too – see Appendix B.3.

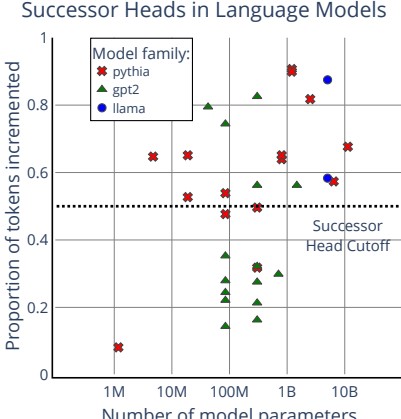 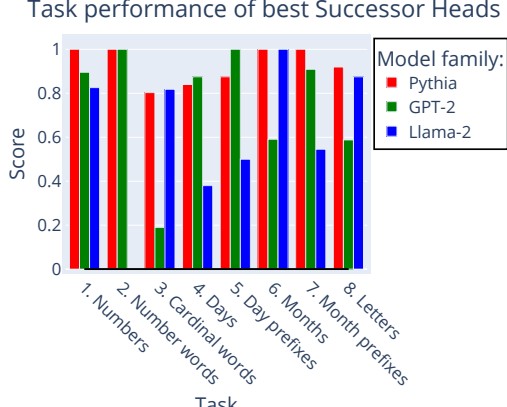

Figure 2: Plots of successor scores (proportion of tokens where succession occurs) for each model tested. A plot of the highest successor score observed across all attention heads for each model tested (left) and successor scores of the best successor heads in models (Pythia-1.4B, GPT-2 XL, Llama-2 7B) across different tasks (right).

Specifically, we use sparse autoencoders to isolate abstract 'mod-10' features within the outputs of the $\text{MLP}_0$ layer (Section 3.1), provide further evidence for these features through linear probing and ablative experiments on individual neurons (Section 3.2), and use these features to steer model behavior across different successor tasks (Section 3.3).

## 3.1 FINDING MOD-10 FEATURES

To uncover more structure within these $\text{MLP}_0$-representations, we train a ***sparse autoencoder*** (SAE) (Ng (2011); Cunningham et al. (2023); Bricken et al. (2023), see Appendix B) on tokens $t$ from a range of ordinal sequences. In short, SAEs find a sparse set of linear features that can reconstruct activations in neural networks. We apply SAEs to the $\text{MLP}_0$-representations of all tokens $t$ and call this their ***SAE-decomposition***. The $i$th SAE feature has activation (on input token $t$) $\alpha_i(t) \geq 0$.

Given an ordinal sequence token $t$ and a trained SAE, we define $t$'s ***most important feature*** $t^\star$ as the SAE feature that, when ablated from the reconstruction of $t$'s $\text{MLP}_0$-representation, causes the biggest decrease in the probability of the successor of $t$ (by calculating probabilities from the logits obtained by multiplying by $W_U W_{OV}$).

For numeric tokens $t_n$ (e.g. $t_{13} = $ '13'), we find that their most important feature is usually shared by other numeric tokens $t_m$ for which $m \equiv n \mod 10$ (e.g. $t_3, t_{23}, ..., t_{93}$). Indeed, for each of the 'mod-10 classes' $\{t_0, ..., t_{90}\}, \{t_1, ..., t_{91}\}, ..., \{t_9, ..., t_{99}\}$, the modal most important feature for tokens in that class is shared on average by $58.5\%$ of tokens in that class. Moreover, we find that the most important feature of $t_i$ typically only has a high activation $\alpha_i(t_j)$ in the SAE-decomposition of $t_j$ if $i \equiv j \mod 10$, which we visualise as 'mod-10 bands' in Figure 3.

We also observe that these features are *causally important* for the successor head to perform incrementation: if we apply the successor head to the most important features $t_i^\star$ of tokens $t_i$ (i.e. compute logits as $W_U W_{OV} t_i^\star$), the resulting distribution places high weight on tokens $t_j$ for $j \equiv i + 1 \mod 10$, which we visualise as 'mod-10 bands' in Figure 5. Furthermore, the weight placed on tokens $t_j$ when $t_i$ is a single-digit number is much larger than that when $t_i$ is a double-digit number.

Given these observations, we hypothesise that the most important features of numeric tokens $t_i$ might in some way encode the value of $i \mod 10$. As such, we define the ***mod-10 features*** $f_0, ..., f_9$ as the modal most important features from mod-10 classes $\{t_0, ..., t_{90}\}, \{t_1, ..., t_{91}\}, ..., \{t_9, ..., t_{99}\}$, averaged over 100 SAE training runs. We verify that these mod-10 features are causally important for incrementation, by observing that the logit distribution obtained by applying the successor head to features $f_i$ (i.e. $W_U W_{OV} f_i$) places high weight on tokens $t_j$ for $j \equiv i + 1 \mod 10$ (which we visualise as 'mod-10 bands' in Figure 4).

## 3.2 TRANSFERABILITY OF MOD-10 FEATURES

Are the mod-10 features we found in Section 3.1 simply an artifact of the SAE technique? We provide evidence that these are natural, causally important features by using two independent methods to recover them: (1) linear probing, and (2) identifying $\text{MLP}_0$ neurons. We also demonstrate that these features transfer to other tasks in the succession dataset (Section 2).

**(1) Linear probing.** We train a linear probe $P : \mathbb{R}^{10 \times d_{\text{model}}}$ to predict the value of $i \mod 10$ from the $\text{MLP}_0$-representations of numeric tokens $t_i$. We find that $P_i$, the $i$th row of our linear probe, has a high cosine similarity (on average 0.70764) to the corresponding mod-10 feature $f_i$ obtained from the SAE. Surprisingly, the probe even generalizes to non-numeric tokens, correctly predicting the index value $\mod 10$ for 94/102 examples from succession dataset tasks 2-8 (Section 2). Our full experimental setup is described in Appendix C.1. Additionally, we provide an analysis of linear probes for moduli other than 10 in Appendix R.

**(2) $\text{MLP}_0$ neurons.** We perform ablative experiments on individual $\text{MLP}_0$ neurons (activations immediately after the final ReLU/GELU) to find the *most important neurons* for incrementing numeric tokens $t_i$ (measured by decrease in probability of the successor token $t_{i+1}$, as per the definition of *most important feature*). Observing the behavior of these neurons on tokens $t_i$ reveals periodic spiking patterns in firing intensity (the neuron's activation value), with the most common period across the top 16 most important neurons being 10. Figure 22 presents the firing patterns of some of these neurons. Indeed, we also find that the neurons increase probability on successor tokens by multiplying the neuron's corresponding direction with $W_U W_{OV}$ in the same figures. Further technical details can be found in Appendix D. Our results on the interpretability of individual neurons may seem surprising in light of recent work suggesting that the individual neurons of language models may be inappropriate as the units of understanding (Elhage et al., 2022). However, our results do not contradict previous findings that understanding MLPs requires understanding distributed behaviors, since, for example, feature $f_6$ appears to be in superposition across at least two neurons (see Figure 22).

## 3.3 VECTOR ARITHMETIC WITH MOD-10 FEATURES

The generalization of our mod-10 linear probe to unseen numeric tasks suggests that token representations across different tasks might be *compositional* and share some common mod-10 structure. In this section, we test our understanding of this structure by performing vector arithmetic with our mod-10 features to manipulate the index of ordinal sequence tokens. For instance, just as Mikolov et al. (2013) found that work *vec*tors satisfied $vec(\text{'King'}) - vec(\text{'Man'}) + vec(\text{'Woman'}) = vec(\text{'Queen'})$, we expect $\text{MLP}_0(W_E(\text{'fifth'})) - k f_5 + k f_7$ (3) to be causally used by the model in a similar way to $\text{MLP}_0(W_E(\text{'seventh'}))$, where $k$ is a scaling factor (Appendix E).

We use our successor head to test this hypothesis. Observe that, if (3) behaves like $\text{MLP}_0(W_E(\text{'seventh'}))$, applying the successor head to (3) (i.e. multiplying by $W_U W_{OV}$) should yield a distribution with more weight on 'eighth' than on any other token from the cardinal-word task in the succession dataset (Section 2). Indeed, this is correct as indicated by the circled checkmark $\oslash$ in Figure 6. We can perform a similar experiment with all tokens in the succession dataset and with features other than $f_7$ added. The cases where the max logits are on the successor token are check-marked in Figure 6. We describe the experiment in more detail in Appendix E and we also display how logits are distributed across tokens for individual cells in Appendix Q. We find that when the mod 10 addition feature is larger than the source value (modulo 10), vector arithmetic works on 53% (for months) and 89% (for digits 20-29) of cases.

**Greater-than bias**. The vector arithmetic experiments (Figure 6) work much worse when the mod-10 addition is smaller than the source tokens's ordinal sequence position mod 10 (e.g. experiments involving 'March' and adding $f_1$ or $f_2$ do not go well). This is because Successor Heads are biased towards values greater than the successor, compared to values less than the successor. This effect can be seen in Figure 7a on the tokens 'first', 'second', ..., 'tenth'. However, our mod-10 features do not exhibit a greater-than bias, as seen in Figure 7b. We survey these effects across all tasks in Appendix I. As a result, using the mod-10 features to shift logits towards tokens of a lower order than the input token fails, as the changes in logits are not significant compared to the large logits on higher-order tokens. In the case of numbers, this leads to the effect that, for exam-

ple, $W_U W_{OV} (\text{MLP}_0(W_E(\text{`35'})) - k f_5 + k f_3)$ has high logits on '43', rather than '33' (this '+10' effect occurs for 2/3 of entries below the diagonal in the 20-29 numbers table of Figure 6).

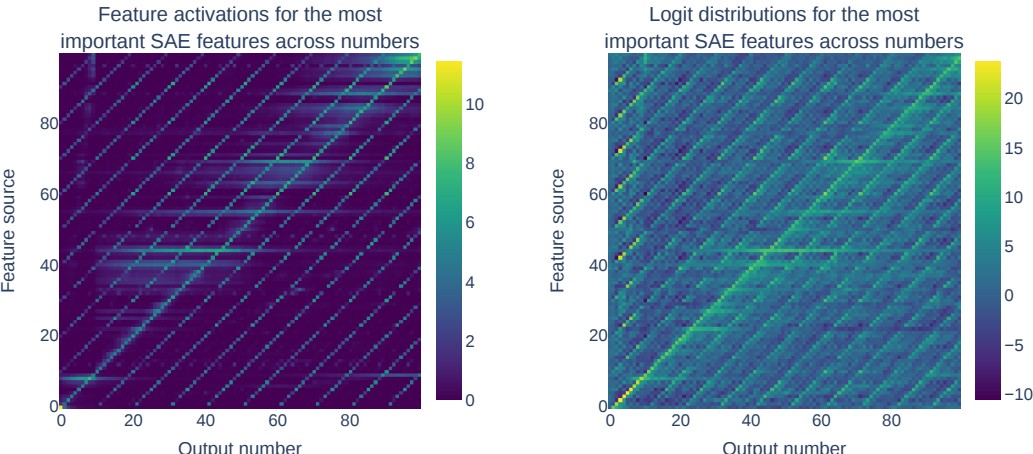

Figure 3: The activations of $t_i$'s most important feature ($y$-axis) in the SAE-decomposition of $t_j$ ($x$-axis), for $t_i, t_j$ numeric tokens. Values averaged over 100 SAE training runs.

Figure 4: The logit value for $t_j$ ($x$-axis) when unembedding the most important feature of $t_i$ ($y$-axis), for $t_i, t_j$ numeric tokens. Values averaged over 100 SAE training runs.

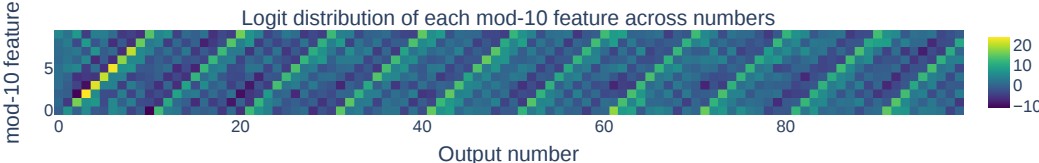

Figure 5: Logit distribution $W_U W_{OV} f_i$ for each mod-10 feature $f_i$.

**Limitations:** The absence of a strong greater-than bias in our mod 10 features suggests this feature-level description is missing some details – specifically, that successor heads must use other numeric information to produce the greater-than bias we observe. Additionally, while we see a good generalization of the mod 10 features across various tasks in the table in Figure 6, the mod 10 features are not able to steer the Days and Letters tasks from Section 2. We describe this in Appendix F.

## 4 SUCCESSOR HEADS IN THE WILD

In this section, we analyze the behaviour of successor heads within natural-language datasets, and observe that they aren't simply responsible for incrementation: indeed, we identify four distinct, interpretable categories of successor head behavior, highlighting successor heads as an example of an ***interpretably polysemantic*** attention head 'in the wild'.

In order to characterize the behavior of Pythia-1.4B's successor head on natural-language data, we *randomly* sample 128 length-512 contexts from *The Pile*, and for each prefix of each context, we assess whether the successor head is important for the model's ability to predict the correct next token. To measure importance, we use ***direct effect mean ablation***, which involves patching the output of a head with its mean output over a chosen distribution (in this case, the same batch), and, at the very end of the model, subtracting this mean output and adding the original head output to the residual stream (other effects and ablation methods are explored in Appendix M). We evaluate prefixes using two different metrics for per-prompt successor head importance:

**Winning cases.** We identify prefixes where the head that most decreases the logit for the correct next token under direct effect mean ablation is the successor head, denoting them as ***winning cases***.

| | 0 | 1 | 2 | 3 | 4 | 5 | 6 | 7 | 8 | 9 |
|---|---|---|---|---|---|---|---|---|---|---|
| 20 | ✓ | ✓ | | ✓ | ✓ | ✓ | ✓ | ✓ | ✓ | ✓ |
| 21 | | ✓ | ✓ | ✓ | ✓ | | | | ✓ | ✓ |
| 22 | | | ✓ | ✓ | ✓ | ✓ | ✓ | ✓ | ✓ | ✓ |
| 23 | | | | ✓ | ✓ | ✓ | ✓ | | ✓ | ✓ |
| 24 | | | | | ✓ | ✓ | ✓ | ✓ | ✓ | ✓ |
| 25 | | | | | | ✓ | ✓ | ✓ | ✓ | ✓ |
| 26 | | | | | | | ✓ | ✓ | | ✓ |
| 27 | | | | | | | | ✓ | | |
| 28 | | | | | | | | | ✓ | ✓ |
| 29 | | | | | | | | | | ✓ |
| ten | ✓ | ✓ | ✓ | ✓ | ✓ | ✓ | ✓ | ✓ | ✓ | |
| eleven | | ✓ | ✓ | ✓ | ✓ | ✓ | ✓ | ✓ | ✓ | ✓ |
| twelve | | | ✓ | ✓ | ✓ | ✓ | ✓ | | ✓ | |
| thirteen | | | | ✓ | ✓ | ✓ | ✓ | ✓ | ✓ | ✓ |
| fourteen | | | | | | ✓ | ✓ | ✓ | ✓ | ✓ |
| fifteen | | | | | | ✓ | ✓ | ✓ | | ✓ |
| sixteen | | | | | | | ✓ | ✓ | ✓ | ✓ |
| seventeen | | | | | | | ✓ | ✓ | ✓ | ✓ |
| eighteen | | | | | | | | ✓ | ✓ | ✓ |
| nineteen | | | | | | | | ✓ | ✓ | ✓ |
| twenty | | | | | | | | ✓ | | ✓ |

| | 1 | 2 | 3 | 4 | 5 | 6 | 7 | 8 | 9 | 0 |
|---|---|---|---|---|---|---|---|---|---|---|
| January | ✓ | ✓ | ✓ | | | | | | | |
| February | | ✓ | ✓ | ✓ | | | | | | |
| March | | | ✓ | ✓ | ✓ | | | | | |
| April | | | ✓ | ✓ | ✓ | | | | | |
| May | | | | ✓ | ✓ | ✓ | | | | |
| June | | | ✓ | | ✓ | ✓ | ✓ | ✓ | ✓ | |
| July | | | | | | | ✓ | ✓ | ✓ | |
| August | | | | | | | | ✓ | ✓ | |
| September | | | ✓ | | | | | | ✓ | ✓ |
| first | ✓ | ✓ | ✓ | ✓ | | | | | | |
| second | | ✓ | ✓ | ✓ | ✓ | | | | | |
| third | | | ✓ | ✓ | ✓ | ✓ | ✓ | | | |
| fourth | | | | ✓ | ✓ | ✓ | ✓ | | | |
| fifth | | | | | | ✓ | ✓ (circled) | ✓ | | |
| sixth | | | | | ✓ | ✓ | ✓ | ✓ | | |
| seventh | | | | | | ✓ | ✓ | ✓ | | |
| eighth | | | | | | | ✓ | ✓ | ✓ | |
| ninth | | | | | | | | | ✓ | |

Example: The circled case (3) has a source token of 'fifth' and a target residue of 7, and it is ticked because vector arithmetic leads to this token being successfully treated like 'seventh')

Figure 6: Table displaying cases in which vector arithmetic such as (3) is successful for various ranges of tokens. Other ranges of tokens give similar results, as displayed in Appendix P. Rows: source tokens. Columns: target residues modulo 10.

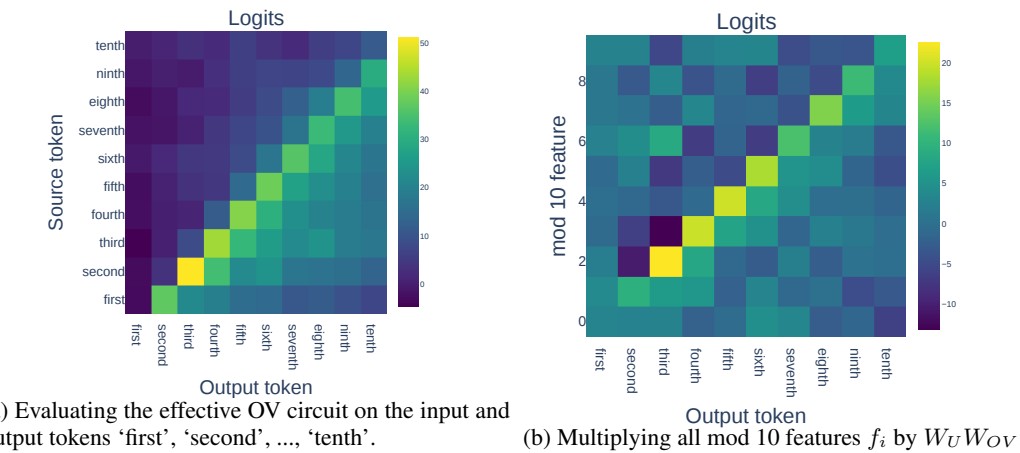

(a) Evaluating the effective OV circuit on the input and output tokens 'first', 'second', ..., 'tenth'.

(b) Multiplying all mod 10 features $f_i$ by $W_U W_{OV}$.

Figure 7: The Successor Head OV circuit displays a clear bias against decrementation (Figure 7a), i.e. the logits on or above the main diagonal are less than the logits below the main diagonal. This bias isn't captured in the mod 10 feature (Figure 7b).

**Loss-reducing cases.** We identify prefixes p where direct effect mean ablation of the successor head increases next-token prediction loss (by $\Delta\mathcal{L}(\mathrm{p})$), denoting them as ***loss-reducing cases***.

### 4.1 INTERPRETABLE POLYSEMANTICITY IN SUCCESSOR HEADS

On analyzing prefixes where the successor head is particularly important for next-token prediction – i.e. loss-reducing and winning cases – we observe four main categories of behavior, which we operationalize as follows (denoting a ***top-$n$-attended token*** as a token at one of the top $n$ positions to which the successor head attends most strongly):

*Successorship behavior*: *the successor head pushes for the successor of a token in the context.* We say this behavior occurs when one of the top-5-attended tokens is in the successorship dataset, and the correct next token is the successor of $t$.

*Acronym behavior:* *the successor head pushes for an acronym of words in the context.* We say this behavior occurs when the correct next token is an acronym whose last letter corresponds to the first letter of the top-1-attended token. (For example, if the successor head attends most strongly to the token 'Defense', and the correct next token is 'OSD'.)

*Copying behavior:* *the successor head pushes for a previous token in the context.* We say this behavior occurs when the correct next token $t$ has already occurred in the prompt, and token $t$ is one of the top-5-attended tokens.

*Greater-than behavior:* *the successor head pushes for a token greater than a previous token in the context.* We say this behavior occurs when we do not observe successorship behavior, but when the correct next token is still part of an ordinal sequence and has greater order than some top-5-attended token (e.g. if the successor head attends to the token 'first' and the model predicts the token 'third').

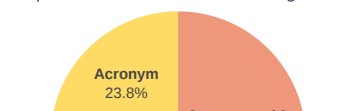

Figure 8: Proportions of three dominant behaviors across winning cases.

| Prompt | Completion |
|---|---|
| (...) [@B2] Hence, bonding to ceramic requires strict attention to detail for optimal clinical outcomes.[@B | 3 |
| (...) designated as boxazomycin A and | B |
| (...) called Generalized Single Step Single Solve ( | GS |
| (...) More than two- | thirds |
| (...) where one or more access points ( | AP |

Figure 9: Random sample of 5 winning cases. Negligibly many winning cases were copying.

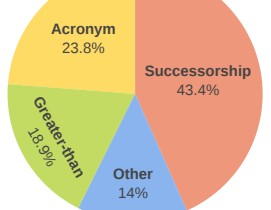

Figure 10: Proportions of reduced loss ($\Delta\mathcal{L}$) attributable to prompts of each behaviour.

| Prompt | Completion |
|---|---|
| (...) low-dose administration of 14- and | 15 |
| (...) in the second round, let's get it to the | third |
| (...) for 1) investigators to ask the missionaries, | 2 |
| (...) You are using Microsoft Test Manager ( | MT |
| (...) known as the Medical Research Council Technology ( | MR |

Figure 11: The top 5 most loss reducing examples.

We plot the proportions of each behavior observed across winning cases in Figure 8, and the fraction of total reduced loss over all contexts ($\Delta\mathcal{L}$) attributable to contexts of each behavior in Figure 10. We also illustrate a random sample of 5 winning cases in Figure 9, and of 5 loss-reducing cases in Figure 11. We observe that, while successorship is the predominant behavior across both winning and loss-reducing cases, acronym and greater-than behaviors also form a non-negligible fraction of successor head behavior. In other words, the successor head is an example of an attention head with ***interpretable polysemanticity***[5]. While polysemanticity has been observed in both vision models (Olah et al., 2020) and toy models trained to perform simple tasks (Elhage et al., 2022), to the best of our knowledge the presence of both successorship and acronym behavior in head L12H0 is the cleanest example of polysemantic behavior identified so far in an LLM, where we show two clear distinct behaviors one model component has. Finally, this finding is surprising given research into polysemanticity and superposition (Elhage et al., 2022; Bricken et al., 2023). The succession and acronym behaviors are different tasks that L12H0 completes, yet they are not independent tasks

---

[5]A component of a network is said to be *interpretably polysemantic* if it performs multiple distinct, interpretable functions.

occur in completely different contexts (this is because a token completion could involve both succession and an acronym, e.g 'the First Limited Corporation, ' could be completed with ' Second' or ' FLC').

Note that in this section, while we identified that successor heads are often used in tasks involving incrementation, we did not explicitly demonstrate that successor heads are *necessary* for incrementation. In Appendix K we describe an experiment that reveals that successor heads are necessary for a specific incrementation task (*numbered listing*).

## 5   RELATED WORK

**Mechanistic Interpretability** research aims to reverse engineer trained neural networks analogously to how software binaries can be reverse-engineered (Olah, 2022). This research was largely developed in vision models (Bau et al., 2017; Olah et al., 2017) though most recent research has studied language models (Elhage et al., 2021; Olsson et al., 2022; Gurnee et al., 2023) and transformers (Nanda et al., 2023). Olah et al. (2020) introduces the universality hypothesis and we use Chughtai et al. (2023)'s 'weak universality' notion in this work (Section 1).

**Transformer circuits**. More specifically, our work builds from the insights of Elhage et al. (2021)'s framework for understanding circuits in transformers, including how autoregressive transformers have a residual stream. Due to the residual stream, different paths from input to output can bypass as many attention heads and MLPs as necessary. This has further been explored in specific case studies (Wang et al., 2023; Goldowsky-Dill et al., 2023) and generalizes to backwards passes (Du et al., 2023). One related case study to our work is Hanna et al. (2023) which studies a Greater-Than circuit in GPT-2 Small, similar to how we indirectly found the Greater-Than operation in Section 3. Hanna et al. (2023) focus mainly on numbers, not other tasks. Our work is inspired by Olsson et al. (2022) who study induction heads and find that heads with similar attention patterns exist in larger models. In our work we provide an end-to-end explanation of generalizing language model components (Figure 1), though induction heads are related to in-context learning and a consistent phase change in training, which we didn't observe for successor heads (Appendix H).

**LLMs and vector arithmetic**. Mikolov et al. (2013)'s seminal work on word embedding arithmetic showed that latent language model representations had compositionality, e.g. $vec$('King') $-$ $vec$('Man') $+ vec$('Woman') $\approx vec$('Queen'). Recently Merullo et al. (2023) showed some extension of these arithmetic results to LLMs. L (2023) found that 'one is 1' and that similar heads in GPT-2 Small boost successors numbers, months and days, which we generalize to more architectures and to an end-to-end circuit (Figure 1). Lan & Barez (2023) also use an automated approach to study the overlap of these tasks. Finally, Subramani et al. (2022); Li et al. (2023) and Turner et al. (2023) use residual stream additions to steer models. Our work differs in that it considers shallow targeted paths through networks, rather than deep hidden states in networks.

## 6   CONCLUSION

In this work, we discovered and interpreted a class of attention heads we call *successor heads*. We showed that these heads increment tokens in ordinal sequences like numbers, months, and days, and that the representations of the tokens are *compositional* and contain *interpretable, abstract 'mod-10' features*. We provided evidence that successor heads exhibit a weak form of universality, arising in models across different architectures and scales (from 31M to 12B parameters), and using similar underlying mod-10 features in all cases. Finally, we validated our understanding by demonstrating that a successor head reduced the loss on training data by predicting successor tokens.

Additional numeric representation findings relevant to future work include:

1. Finding a 'greater-than bias', where a language model was much more likely to predict numeric answers larger than the values in the prompt, compared to smaller values than tokens present in the prompt, that was observable by a weights-level analysis.
2. Surprisingly interpretable individual MLP0 neurons on this narrow task.
3. A novel example of attention head polysemanticity (successor heads predicting acronyms).

## 7 ACKNOWLEDGEMENTS

We would like to thank Bilal Chughtai, Théo Wang and reviewers for comments on a draft of this work and Neel Nanda for a helpful discussion, as well as Lawrence Chan and Sebastian Farquhar for pieces of advice. Elizabeth Ho, Will Harpur-Davies and Andy Zhou worked on an early version of this work in GPT-2 Small with help from Théo Wang.

Contributions from each author: Rhys Gould first found successor heads in GPT-2 Small and Pythia-1.4B, and identified the mechanism from Figure 1 as well as the mod-10 features. Euan Ong developed Appendix A, improved our understanding, and worked on writing across the paper. George Ogden found successor heads in larger models (e.g Figure 2). Arthur Conmy led the project, framed the paper's contributions and suggested and implemented several experiments throughout the work.

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

## A  ORDINAL SEQUENCES ARE REPRESENTED COMPOSITIONALLY

Let $i_s$ denote the $i$th token in ordinal sequence $s$ (such that e.g. $2_{\text{Month}}$ corresponds to the token 'February'), and let $[\![i_s]\!] = \text{MLP}_0(W_E(i_s))$ denote the model's internal ***MLP$_0$-representation*** of token $i_s$ (the output of MLP$_0$ in Figure 1).

Given successor heads $S = W_{OV}$ can increment tokens $s_i$ from a range of ordinal sequences $s$ (e.g. numbers, months, days of the week), one might hypothesise that the MLP$_0$-representations of such tokens have *compositional structure* – i.e. that information about a token's *position* $i$ in its ordinal sequence is encoded independently from information about *which ordinal sequence* $s$ it comes from. More precisely, we claim that we can decompose representations $[\![i_s]\!]$ into features $\mathbf{v}_i$ living in some 'index space' and $\mathbf{v}_s$ living in some 'domain space', such that $[\![i_s]\!] = \mathbf{v}_i + \mathbf{v}_s$.

**Method.** To test this compositionality hypothesis, we wish to learn two linear maps – an *index-space projection* $\pi_{\mathbb{N}} : \mathbb{R}^{d_{\text{model}}} \to \mathbb{R}^{d_{\text{model}}}$ and a *domain-space projection* $\pi_D : \mathbb{R}^{d_{\text{model}}} \to \mathbb{R}^{d_{\text{model}}}$ – such that, for all pairs of tokens $i_s$ and $j_t$ (with $i_t$ a valid token), $[\![\hat{i_t}]\!] := \pi_{\mathbb{N}}([\![i_s]\!]) + \pi_D([\![j_t]\!]) \approx [\![i_t]\!]$. To do so, we enforce that $\pi_{\mathbb{N}} + \pi_D = I$, and ensure predicted representations $[\![\hat{i_t}]\!]$ 'behave like' ground truth representations $[\![i_t]\!]$ for randomly sampled pairs of tokens $i_s$ and $j_t$ – in other words, that there is low $L_2$-distance between $[\![\hat{i_t}]\!]$ and $[\![i_t]\!]$, that $[\![\hat{i_t}]\!]$ decodes to $i_t$ (***output-space decoding***), and that $S([\![i_t]\!])$ decodes to $(i+1)_t$ (***successor decoding***). For full experimental details see Appendix N.

| | | Source index token ($i_{\text{Rom}}$) | | | | | | | | | | |
|---|---|---|---|---|---|---|---|---|---|---|---|---|
| | | I | II | III | IV | V | VI | VII | VIII | IX | X | XI | XII |
| | **1** | 9 | 2 | 3 | 4 | 22 | 6 | 7 | 8 | 9 | 24 | 11 | 12 |
| | **one** | nine | two | three | four | twenty | six | seven | eight | nine | twenty | eleven | twelve |
| | **first** | ninth | second | third | fourth | second | sixth | seventh | eighth | ninth | fourth | - | - |
| Sequence token ($1_s$) | **Monday** | Sunday | Tuesday | Wedne... | Thursday | Tuesday | Saturday | Sunday | - | - | - | - | - |
| | **Mon** | Sep | Tue | Wed | Thu | Tue | Sat | Sun | - | - | - | - | - |
| | **January** | Septem... | February | March | April | February | June | July | August | Septem... | Decem... | Novem... | Decem... |
| | **Jan** | Sep | Feb | Mar | Apr | Feb | Jun | Jul | Aug | Sep | Dec | Nov | Dec |
| | **A** | I | B | C | D | V | F | G | H | I | X | W | L |

Table 2: A table presenting top-1 predicted tokens under output-space decoding from $\pi_D(1_s) + \pi_{\mathbb{N}}(i_{\text{Rom}})$. Green cells denote predictions which match their target exactly; red cells denote incorrect predictions. Dashed cells denote pairs of $1_s$ and $i_{Rom}$ for which $i_s$ is not a valid (single) token.

**Results.** On our held-out dataset of token pairs, we obtained a top-1 output-space decoding accuracy of 1.00. To explore out-of-distribution performance, we also test whether $\pi_{\mathbb{N}}$ can project out the numeric component of Roman numerals (which weren't in the successor dataset), by taking Roman numerals $i_{\text{Rom}} \in \{\text{'I'}, ..., \text{'XII'}\}$ and tokens $1_s$ from sequences $s$ in the successor dataset, and testing whether $\pi_D(1_s) + \pi_{\mathbb{N}}(i_{\text{Rom}})$ decodes to $i_s$. We present the top-1 predicted tokens under output-space decoding in Table 2: observe that we obtain perfect top-1 accuracy (apart from $i \in \{1, 5, 10\}$, which we can attribute to the Roman numerals I, V and X being single-letter and impossible to disambiguate from $9_{\text{Letter}}$, $22_{\text{Letter}}$ and $24_{\text{Letter}}$).

These results – in particular, our ability to project the numeric component out of tokens from unseen sequences and transfer indices across domains – suggest that there is a shared numeric subspace storing the index of a token within its ordinal sequence. Indeed, informal testing suggests that this numeric subspace may be interpretable even for tokens not part of an ordinal sequence: for instance, $d(\pi_{\mathbb{N}}(\llbracket \text{' triangle'} \rrbracket) + \pi_D(1_{\text{Num}}))$ yields $3_{\text{Num}}$, and $d(\pi_{\mathbb{N}}(\llbracket \text{' week'} \rrbracket) + \pi_D(1_{\text{Num}}))$ yields $7_{\text{Num}}$.

We note in Appendix O, however, that applying the successor head to these learned representations did not always preserve performance (i.e. for a constructed representation $\llbracket \hat{i_t} \rrbracket$, $S(\llbracket \hat{i_t} \rrbracket)$ did not always decode to $(i + 1)_t$). This suggests our numeric projection $\pi_{\mathbb{N}}$ might be capturing slightly more than just the numeric subspace: specifically, there may be some components of domain-space which are ignored by output-space decoding, but which our successor head lifts into output-space.

# B  SPARSE AUTO-ENCODERS

## B.1  DEFINITION

We refer to a single-layer autoencoder with a sparsity regularization term in its loss as a **sparse auto-encoder**.

For a dataset generated from a set of underlying vectors (each dataset example is a sparse linear combination of such vectors), it has been empirically observed (Sharkey et al., 2022; Cunningham et al., 2023) that sparse auto-encoders are capable of retrieving the underlying set of vectors. We hope to obtain a set of sparse, interpretable features from the SAEs that decompose some of the structure of MLP$_0$ space that we can use to analyze the way numeric operations are performed.

## B.2  TRAINING PROCESS FOR MOD 10 FEATURES

Training a sparse auto-encoder with $D$ features and regularization coefficient $\lambda$ on a dataset of tokens in MLP$_0$ space results in a map $F : \text{Tokens} \rightarrow (\mathbb{R}^d \times \mathbb{R}_+)^D$, with $F(x) = \{(\mathbf{v}_1, a_1), \ldots, (\mathbf{v}_D, a_D)\}$, mapping a token to a set of feature and feature-activation pairs, with reconstruction $R(x) = \sum_{i=1}^{D} a_i \mathbf{v}_i$. Each $\mathbf{v}_i$ the ReLU of a linear transformation with the input to the SAE, represented by $W_e$ in Figure 12. Note that we use SAEs in MLP$_0$ space, i.e. the reconstruction loss is at the middle stage of Figure 1 which we have illustrated in Figure 12.

We train the SAE using number tokens from 0 to 500, both with and without a space ('123' and ' 123'), alongside other tasks, such as number words, cardinal words, days, months, etc. 90% of

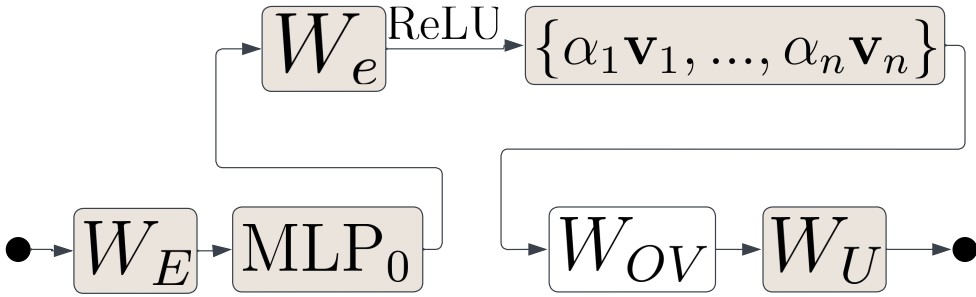

Figure 12: SAEs are trained on the activation after $\text{MLP}_0$ and before $W_{OV}$.

these tokens go into the train set, and the remaining 10% to the validation set. Even with the other tasks, the dataset is dominated by numbers, but creating a more balanced dataset would give us less data to work with, and without enough data, the SAE fails to generalize to the validation set. Hence, we only concern ourselves with the features that the SAE learns for number tokens, and we then separately check whether these features generalize to the other tasks on the basis of logits, rather than SAE activations.

We used the hyperparameters $D = 512$ and $\lambda = 0.3$, with a batch size of 64, and trained for 100 epochs. To find these hyperparameters, we used the metric of mean max cosine similarity between two trained SAEs, as described in Sharkey et al. (2022) and Cunningham et al. (2023).

### B.3 UNIVERSALITY OF MOD-10 RESULTS

We also observe the mod 10 structure via SAEs across models other than Pythia-1.4B, without any finetuning of SAE parameters to these models. We reproduce the SAE figures seen in Section 3.1 for other models, with Appendix B.3 for Pythia-2.8B, and Appendix B.3 for celebrimbor-gpt2-medium-x81.

## C TEST SET EVALUATION

### C.1 LINEAR PROBING

We train a linear probe to predict the mod 10 value of tokens. Specifically, we train on number tokens from '0' to '500', both with and without a space, assigning 90% of tokens to a train set, and the remaining 10% to a validation set. We use a learning rate of 0.001, and a batch size of 32, for 100 epochs.

We then evaluate on a dataset of unseen tasks, including number words (from 'one' to 'nineteen'), placements, Roman numerals, months, days, and any valid spaced and capitalized variants. Out of the total 102 such examples, 94/102 are correct, and the 8 failures are: ['January', 'December', 'Friday', 'Saturday', 'Sunday', ' V', ' X', ' XV'].

The failures of 3 out of 7 days are consistent with our inability to interpret the day task well with our mod 10 features. Additionally, we see 'January' and 'December' as failure cases, which is also consistent with our finding that there does not seem to be a mod 10 feature that corresponds to any of them: $f_1$ behaves as 'November' rather than 'January', and $f_2$ as 'February'.

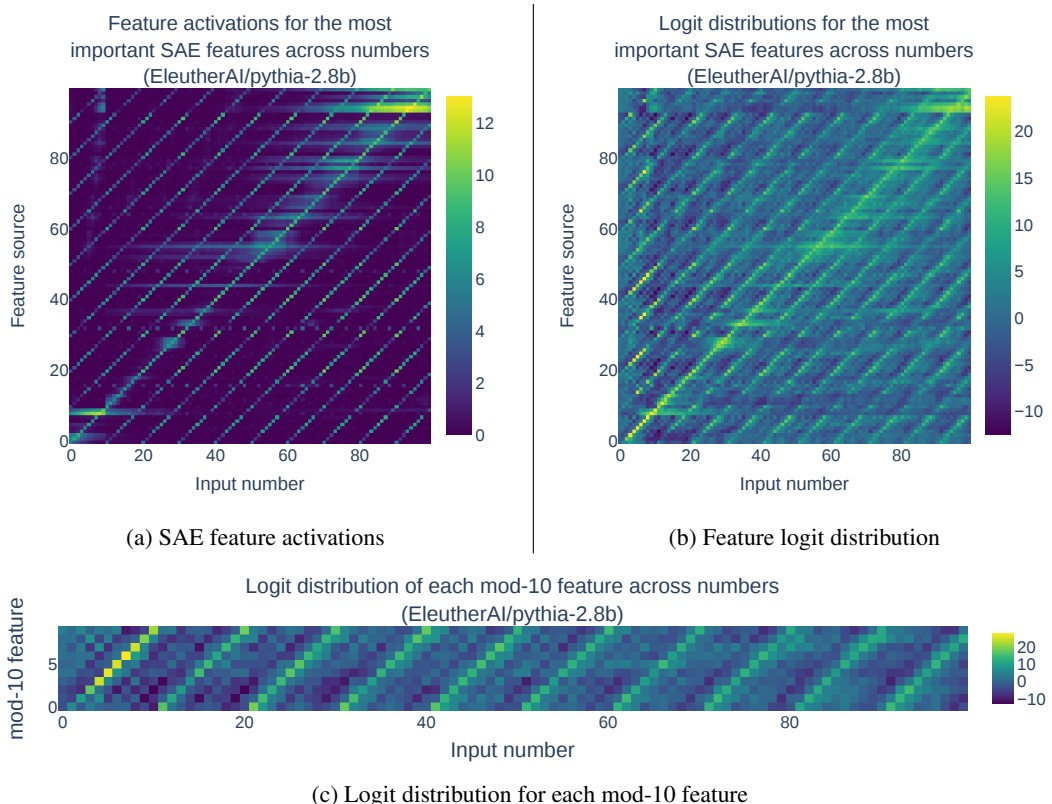

(a) SAE feature activations

(b) Feature logit distribution

(c) Logit distribution for each mod-10 feature

Figure 13: SAE plots for Pythia-2.8B analogous to Figure 3, Figure 4, and Figure 5.

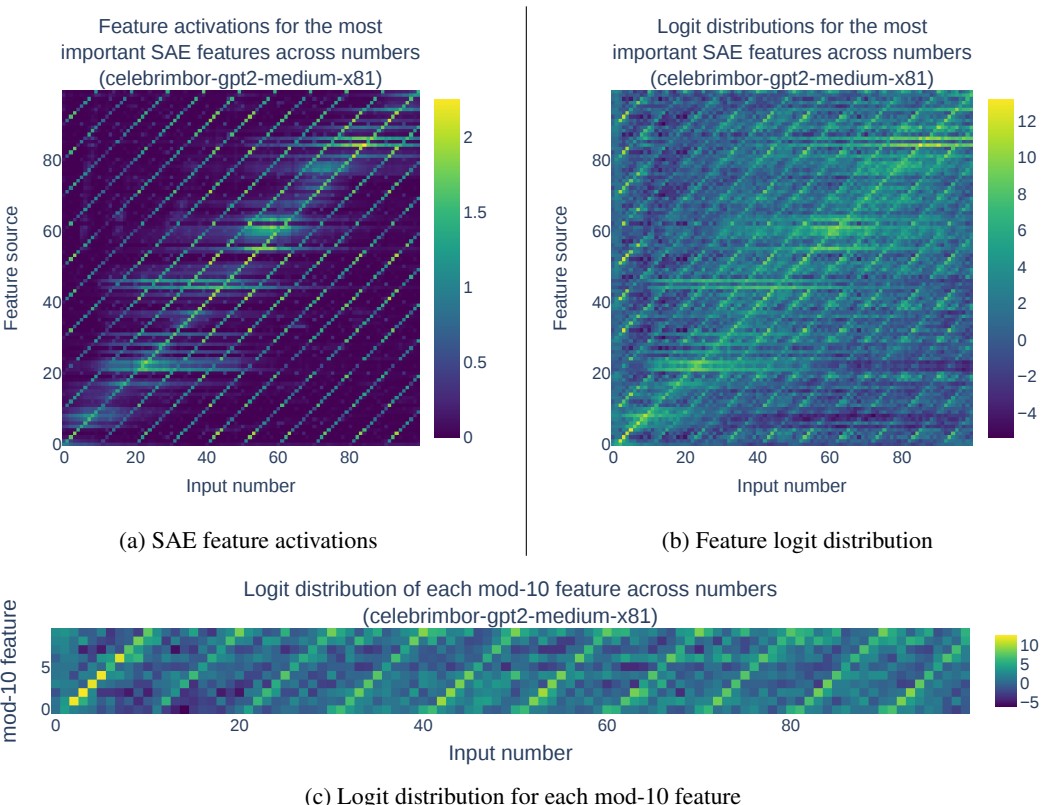

(a) SAE feature activations

(b) Feature logit distribution

(c) Logit distribution for each mod-10 feature

Figure 14: SAE plots for celebrimbor-gpt2-medium-x81 analogous to Figure 3, Figure 4, and Figure 5

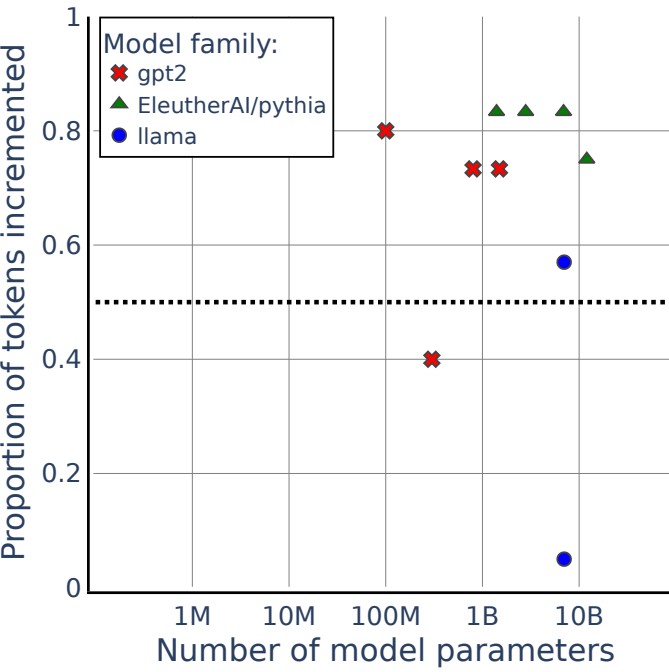

In this work, we used all tokens in numeric sequences in the models' vocabularies, except Roman numerals, so can use these as a test set, as in Appendix C.1 and Appendix N. We tested all the OpenAI GPT-2 models as well as the Pythia models with at least 1B parameters. We find that Successor Heads have variable performance on this held out task, with many (including all Pythia models) achieving a high succession score. However, the original Llama-7B does not generalize well to this task.

Figure 15: (Left) Succession scores *for the Roman numerals task only.*

## C.2 HELD-OUT TASK

## D MLP$_0$ NEURONS

In our MLP$_0$ neuron experiments, we do the following: for each $T \in \{\text{'0', '1', \dots, '99'}\}$, we ablate each neuron from the final activation in MLP$_0$ (the final activation is just before the final linear layer of MLP$_0$), and store the probability attributed to the successor of $T$ after passing the modified (due to ablation) MLP$_0$ output through the successor.

Averaging the correct probabilities across all 100 prompts then gives an averaged correct probability for each neuron after ablation. We then look at the intensities (neuron activation values) and logits across inputs of number tokens for neurons with the lowest correct probability after ablation, meaning they have the most impact on successorship when ablated. This gives us the plots seen in Figure 22.

## E ARITHMETIC EXPERIMENTS

For a token $t$ (row of arithmetic table) with order $n := \text{ord}(t)$, and mod-10 feature $f_i$ (column of arithmetic table), we consider how $x := \text{MLP}_0(W_E(t)) + k(-f_n + f_i)$ attributes logits to tokens within the same task as $t$, with $k \in \mathbb{R}_+$ a scaling. We denote whether $x$ correctly attributes maximal logits to the successor token $t^+$ of $t$ (defined by the property that $\text{ord}(t^+) = n + 1$) by a checkmark, as seen in Figure 6. Since the mod-10 features $\{f_i\}_i$ obtained from the SAE are normalized to unit norm, the scaling $k$ is necessary in order to modify the order of a numeric token. For example, though $f_1$ may be present in the MLP$_0$ embedding of ' eleven', we do not know the *intensity* of the feature (analogous to SAE feature strengths $\{\alpha_i(t)\}_i$) in the embedding. We use the heuristic of a scaling $k := \lambda(\text{MLP}_0(W_E(t)) \cdot f_n \in \mathbb{R}$ for some $\lambda \in \mathbb{R}_+$. The appropriate $\lambda$ should be such that performing the arithmetic described by $x$ has an effect on the embedding's numeric order (i.e. $\lambda$ large enough) while not corrupting the task identity of the original embedding (i.e. $\lambda$ not too large), which is checked by multiplying $x$ by $W_U W_{OV}$ and observing whether the top token has these two properties (that is, whether the numeric order of the token has been altered while maintaining the same task identity of, say, a month). By checking this criterion for a range $\lambda \in$

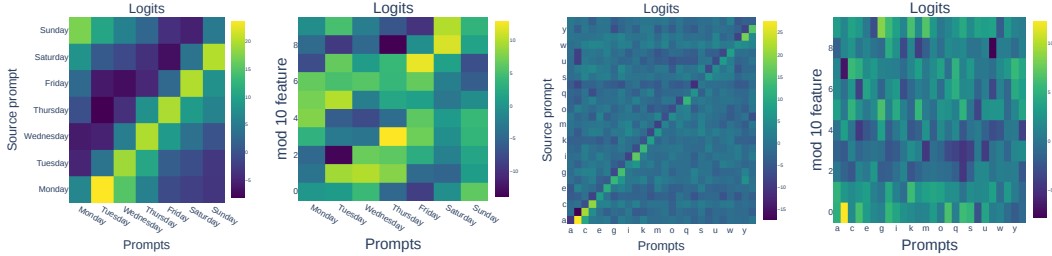

(a) Output logits for successor heads on day tokens (cf. Fig. 7a)

(b) Presence of mod-10 Features on Day Token Predictions (cf. Fig. 7b)

(c) Output logits for successor heads on letter tokens (cf. Fig. 7a)

(d) Presence of mod-10 features on letter tokens (cf. Fig. 7b)

Figure 16: Comparative analysis of output logits for day and letter tasks and presence of mod-10 features. This complements the main text analysis in Figures 7a and 7b, limitations of mod-10 features across certain tasks successor heads perform.

$\{0.0, ..., 0.75, 1.0, 1.25, ..., 3.0\}$, we find that $\lambda = 1$ achieves this criterion for all tasks other than months, where months instead have $\lambda = 2$.

## F  FAILURE CASES OF MOD 10 FEATURES

For the day and alphabet task, analogously to Figure 7, we look at the logits across the task and the mod 10 features. These are displayed in Figure 16, and demonstrate that our mod 10 features are not very interpretable in the context of days and the alphabet in terms of logits, with no clear diagonal of high logits.

## G  RESIDUAL CONNECTIONS ARE NOT IMPORTANT FOR SUCCESSION

To show that there is no relevant information in the residual stream, i.e. the path $W_U \text{MLP}_0(W_E)$ is not sufficient to predict successors, we perform an experiment using the Tuned Lens (Belrose et al., 2023), which approximates the optimal predictions after a given layer inside a transformer.

For all tasks in the succession dataset (Section 2), we used prompt formats (where | denotes a gap between tokens)

1. `|Here| is| a| list|:| alpha| beta| gamma| and| here| is| another|:|<token1>|`
2. `|The|Monday|Tuesday|Wednesday| and| The|<token1>|`

in order to measure how well models were able to predict the successor `<token2>` (e.g 'February') given the final token of the prompt was `<token1>` (e.g 'January'), as LLMs, predict successors given these prompts.

We then took GPT-2 Small and Pythia-1.4B's output after $\text{MLP}_0$ and used the Tuned Lens to get logits on output tokens.[6] The resulting successor score was <1% and commonly predicted bigrams, such as `<token1>`=" first" giving " time" as a completion and `<token1>`=' Sunday' giving ' morning' as a prediction. This suggests that $\text{MLP}_0$ information is insufficient for incrementation and the successor head is critical for succession.

## H  TESTING SUCCESSOR SCORE OVER TRAINING STEPS

Another line of evidence that Successor Heads are an important model component for low training loss can be found by studying successor scores across training points. We study a Pythia model (Biderman et al., 2023) as well as a Stanford GPT model (CRFM, 2021), as these models have training checkpoints. The emergence of Successor Heads throughout training is displayed in Figure 17.

---

[6]Note we did run with GPT-2 Small's attention layer 0 to maximise the model's chances are being able to perform succession. Pythia-1.4B has parallel attention so we just take the MLP0 output in this case.

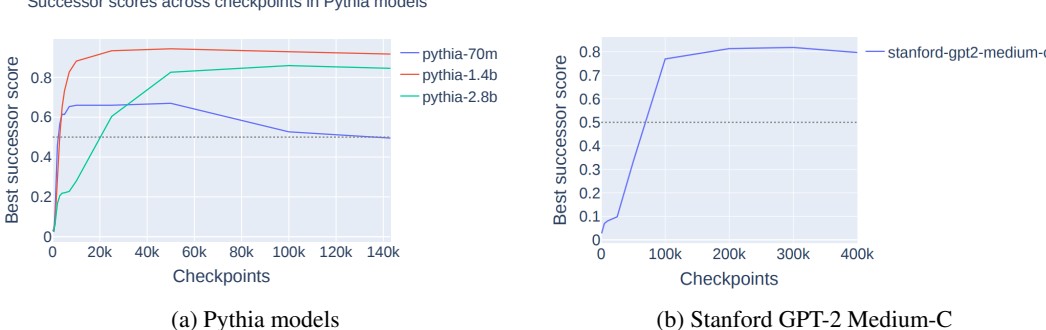

(a) Pythia models

(b) Stanford GPT-2 Medium-C

Figure 17: Best successor scores across successor heads throughout training checkpoints for Pythia and stanford-gpt2 models.

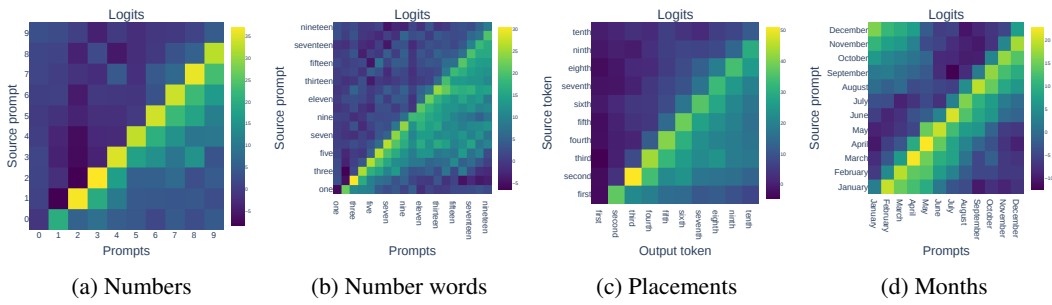

(a) Numbers

(b) Number words

(c) Placements

(d) Months

Figure 18: Plots of logits across various numeric classes, analogous to Figure 7a. Source tokens are on the $y$-axis and output tokens are on the $x$-axis.

## I  DECREMENTATION BIAS ACROSS DIFFERENT TASKS

We show the strength of the decrementation bias in figures Figure 18 and 19.

## J  ALL SUCCESSOR SCORES IN A MODEL

In Appendix J we find that for both Pythia-1.4B (the mainline model in the paper) and GPT-2 Large (a randomly selected model without a successor head from (Figure 2, left)), the heads with highest successor score are sparse: in Pythia-1.4B L12H0 has eight times as great a successor score to the next higher successor score.

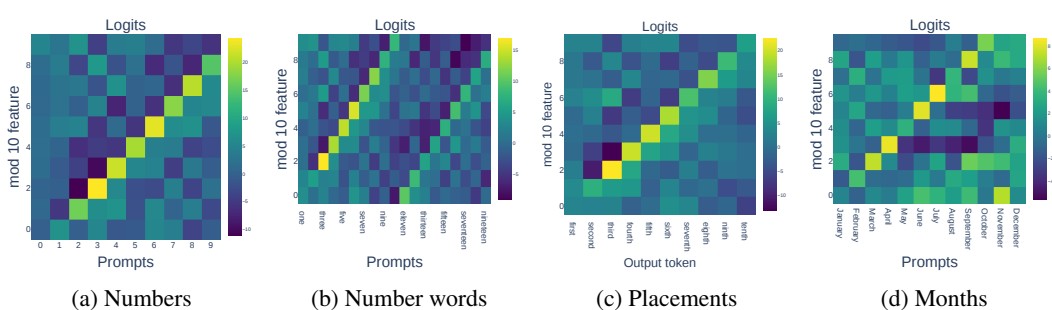

(a) Numbers

(b) Number words

(c) Placements

(d) Months

Figure 19: Plots of mod 10 feature logits across various numeric classes, analogous to Figure 7b

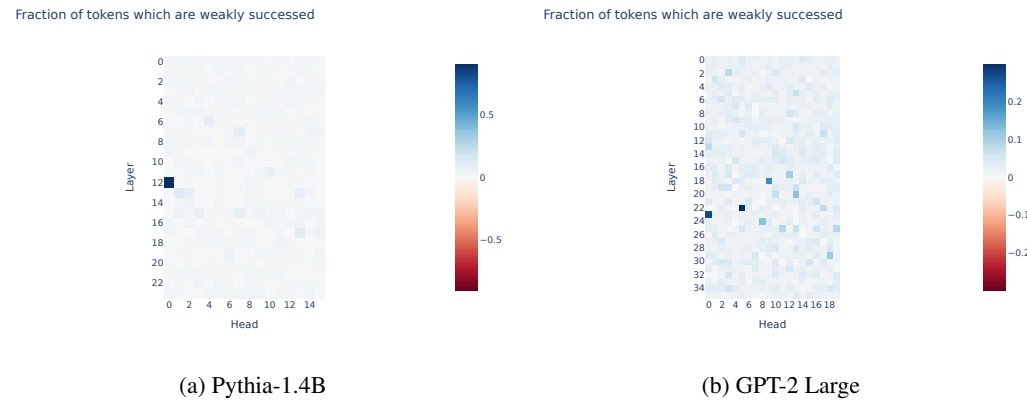

(a) Pythia-1.4B  (b) GPT-2 Large

Figure 20: Successor scores for Pythia-1.4B and GPT-2 Large.

| Prompt | Answer |
|---|---|
| (...) (A) Colony formation and ( < | B |
| (...) (i) $f_g^*(y)$ equals the factual density $f(y)$ for all $g \in \mathbb{G}$; ( | ii |
| (...) [^2]: Conceived and designed the experiments (...) [^ | 3 |
| (...) 6. Kirovsky Zavod Station – St. Petersburg, Russia (...) you can see a statue of Lenin here. | 7 |
| (...) [9] Minutes, Criminal Law Revision Commission, January 28, 1972, 16.[ | 10 |

Figure 21: Some examples of numbered listing prompts from the Pile dataset.

## K  CASE STUDY: NUMBERED LISTING

In Section 4 we demonstrate that when the successor head is contributing usefully, the prompts often required some kind of incrementation. However, we want to investigate whether the converse holds: are prompts requiring incrementation mostly solved by successor heads?

Numbered listing is widespread across real datasets and requires incrementation. Additionally, blog post discussion[7] suggests that even small LLMs are capable of this task in the case of incrementing citations. Examples of prompts involving numbered listing can be seen in Figure 21.

We collect 64 such prompts and check for whether the successor head in Pythia-1.4B is a winning case (as in Section 4), and find that the successor head is indeed the winning head across **all** 64 prompts. Hence this provides some evidence prompts requiring incrementation in real datasets are indeed mostly solved by successor heads.

## L  FIRING PATTERNS OF $\text{MLP}_0$ NEURONS

*Moved to appendix based on suggestions to emphasise technical background more in the main text.*

Firing and logit patterns for the top 16 most important $\text{MLP}_0$ neurons are displayed in Figure 22. We see a superposition effect, with Figure 22c and Figure 22f both representing $6 \mod 10$.

---

[7]https://www.lesswrong.com/posts/LkBmAGJgZX2tbwGKg/
help-out-redwood-research-s-interpretability-team-by-finding

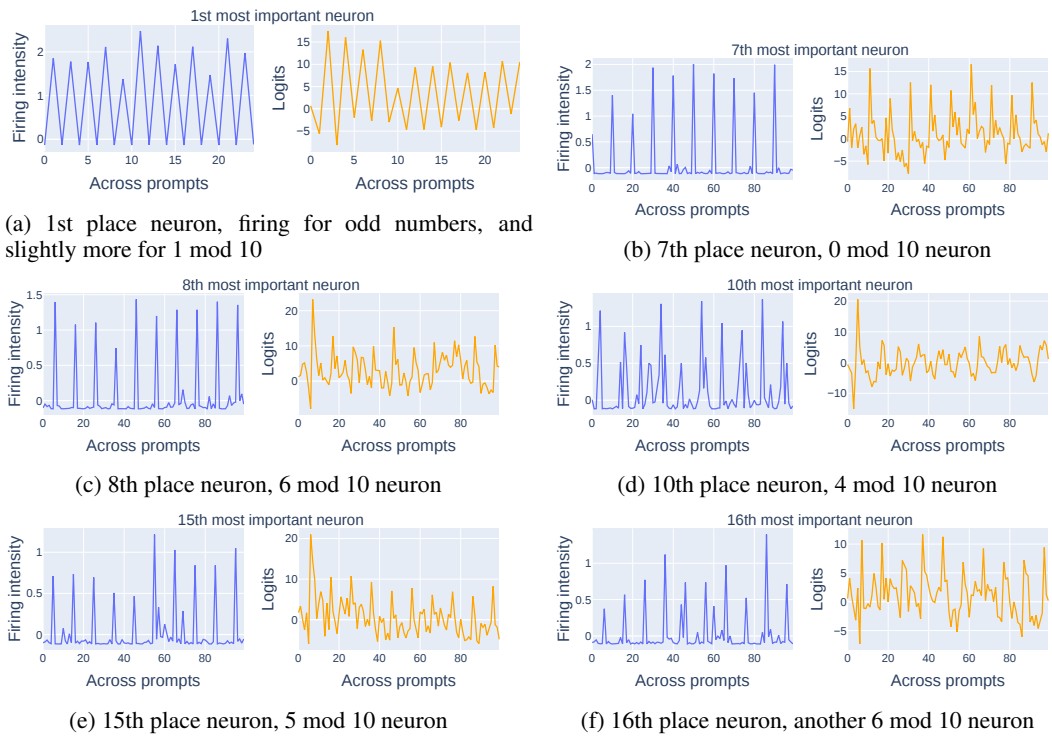

(a) 1st place neuron, firing for odd numbers, and slightly more for 1 mod 10

(b) 7th place neuron, 0 mod 10 neuron

(c) 8th place neuron, 6 mod 10 neuron

(d) 10th place neuron, 4 mod 10 neuron

(e) 15th place neuron, 5 mod 10 neuron

(f) 16th place neuron, another 6 mod 10 neuron

Figure 22: Some examples of neurons firing strongly in modulo 10 patterns out of the top 16 most important MLP0 neurons for successorship.

## M  DIFFERENT ABLATION METHODS

To analyze the effect of language model components when running ablation experiments it is important to distinguish the direct, indirect and total effect of language model components (McGrath et al., 2023; Pearl, 2009) on model outputs, which are illustrated in Figure 23. To measure the direct effect of a component (for a given ablation method) the **direct effect** involves, at the end of the model, subtracting the head's output and adding the ablated output of the head to the residual stream. In Section 4, we analyzed direct effect under the ablation method of mean ablation. This appendix argues that our the direct effect is the largest effect and our results hold under different ablation methods.

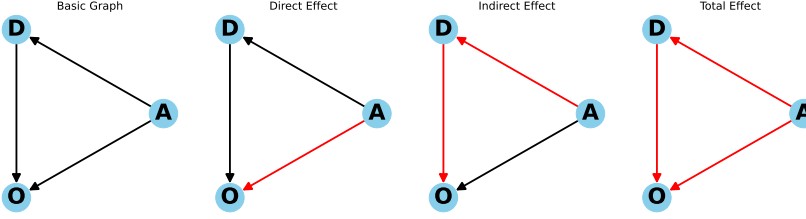

Figure 23: The types of effect an Attention Head (A) could have on model output (O), possibly through mediating downstream model components (D).

The **indirect effect** instead involves replacing a head's output with an ablated output and, at the very end of the model, subtracting this ablated output and adding the head's original output. This effectively ablates the downstream effects of a head.

The ***total effect*** replaces the head's output with an ablated output, which effectively ablates both the direct and indirect effect of the head.

We ablate head outputs using one of the two techniques:

1. Mean ablation: replacing the output of a head with the average head output over a chosen distribution. We choose this distribution to be the current batch we are using.

2. Resampling ablation: replacing the output of a head with the head's activation on a randomly sampled example from a dataset (e.g. the Pile). When using this method, we repeat this process a number of times and average the results.

Rerunning the loss reducing experiments described in Section 4.1 for these different methods gives:

1. Direct effect resampling ablation: 33% for successorship, 11% for acronym, 10% for greater-than, and 11% for copying behaviour.

2. Indirect effect mean ablation: 6.5% for successorship, 2.8% for acronym, 12% for greater-than, and 9.0% for copying behaviour.

3. Indirect effect resampling ablation: 5.8% for successorship, 2.7% for acronym, 11% for greater-than, and 8% for copying behaviour.

4. Total effect mean ablation: 27% for successorship, 8.2% for acronym, 10% for greater-than, and 8.4% for copying behaviour.

5. Total effect resampling ablation: 30% for successorship, 6.5% for acronym, 11% for greater-than, and 6.2% for copying behaviour.

Result 1 demonstrates that using resampling ablation instead of mean ablation for analyzing the loss reducing effect has little effect on the results in Figure 10. Additionally, the indirect effect results (2 and 3) provide some evidence that downstream effects of the successor head are not highly significant to successorship.

## N    TRAINING DETAILS FOR COMPOSITIONALITY EXPERIMENTS

**Remark: obtaining a decoding function.** Recall that we wish to learn two linear maps – an *index-space projection* $\pi_{\mathbb{N}} : \mathbb{R}^{d_{\text{model}}} \to \mathbb{R}^{d_{\text{model}}}$ and a *domain-space projection* $\pi_D : \mathbb{R}^{d_{\text{model}}} \to \mathbb{R}^{d_{\text{model}}}$ – such that, for all pairs of tokens $i_s$ and $j_t$ (with $i_t$ a valid token), $[\![\hat{i_t}]\!] := \pi_{\mathbb{N}}([\![i_s]\!]) + \pi_D([\![j_t]\!]) \approx [\![i_t]\!]$. To evaluate the above identity, we must first learn a decoding function $d : \mathbb{R}^{d_{\text{model}}} \to \mathbb{R}^{d_{\text{vocab}}}$, such that $\arg\max_t d([\![i_s]\!])_t = i_s$. Given the informal observation that directly unembedding $[\![i_s]\!]$ yields next-token predictions for $i_s$ whereas unembedding $S(i_s)$ yields $(i+1)_s$ (see Appendix G), we hypothesise that the unembedding matrix $W_U$ reads from some 'output space' $\mathcal{O}$ and the embedding transform $[\![\cdot]\!]$ writes to some 'input space' $\mathcal{I}$ – and that the successor head reads from $\mathcal{I}$ and writes to $\mathcal{O}$. Indeed, when training an *output-space projection* $\pi_O : \mathcal{I} \to \mathcal{O}$ over tokens in the vocabulary such that $W_U(\pi_O([\![i_s]\!])) = i_s$, we obtain 97.4% top-1 accuracy on a set of 1000 held-out tokens – which both confirms the output-space hypothesis, and gives us a decoding function $d(\mathbf{x}) = W_U(\pi_O(\mathbf{x}))$.

**Method.** With our decoding function in hand, we can train $\pi_{\mathbb{N}}$ and $\pi_D$ to satisfy our identity. Specifically, we define $\pi_{\mathbb{N}}$ and $\pi_D$ to be matrices such that $\pi_{\mathbb{N}} + \pi_D = I$. For valid token pairs $i_s$ and $j_t$, we obtain predicted representations $[\![\hat{i_t}]\!] = \pi_{\mathbb{N}}([\![i_s]\!]) + \pi_D([\![j_t]\!])$, and minimise a combination of 'closeness metrics':

$$||[\![\hat{i_t}]\!] - [\![i_t]\!]||^2 + \mathcal{L}(W_U(\pi_O([\![\hat{i_t}]\!])), i_t) + \mathcal{L}(W_U(S([\![\hat{i_t}]\!])), W_U(S([\![i_t]\!])))$$

for $\mathcal{L}$ the cross-entropy loss. Specifically, we ensure that predicted and ground truth representations 'behave in the same way' – in other words, that they are close together, that predicted representations $[\![\hat{i_t}]\!]$ decode to tokens $i_t$ (***output-space decoding***), and that the logit distribution when decoding incremented predicted representations $S([\![i_t]\!])$ matches that when decoding incremented ground truth representations $S([\![i_t]\!])$ (***successor decoding***).

More succinctly, we can frame the training procedure as learning $\pi_{\mathbb{N}}, \pi_D$ such that the following diagram commutes:

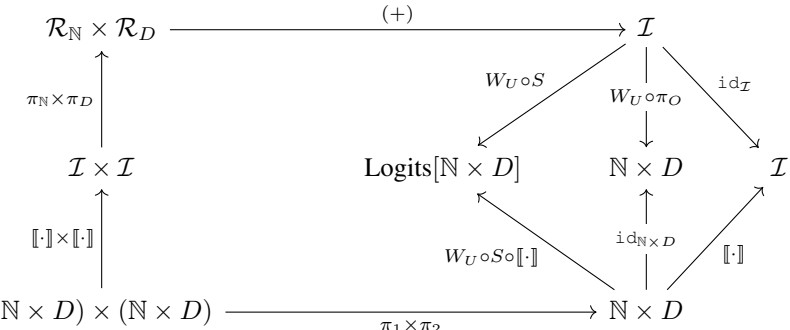

We trained for 10 epochs over valid token pairs sampled from the succession dataset, and evaluated on a held-out dataset of 500 randomly-sampled token pairs.

## O    EFFECTS OF APPLYING THE SUCCESSOR HEAD TO COMPOSITIONAL REPRESENTATIONS

In Section A, we assessed the performance of our factoring mechanism through *output-space decoding* (i.e. evaluating a representation $[\![\hat{i}_t]\!]$ by testing whether $[\![\hat{i}_t]\!]$ decodes to $i_t$). Below, we also present results for *successor-space decoding* (i.e. evaluating a representation $[\![\hat{i}_t]\!]$ by testing whether $S([\![\hat{i}_t]\!])$ decodes to $(i+1)_t$).

In contrast to the high performance of output-space decoding, we found successor-space decoding yielded a top-1 accuracy of 0.90 on the held-out dataset of token pairs, and a top-1 accuracy of 0.125 on the out-of-distribution Roman numeral dataset (see Table 3).[8]

| | | Source index token ($i_{\mathrm{Rom}}$) | | | | | | | | | | | |
|---|---|---|---|---|---|---|---|---|---|---|---|---|---|
| | | I | II | III | IV | V | VI | VII | VIII | IX | X | XI | XII |
| Sequence token ($1_s$) | **1** | 10 | Third | Fourth | 5 | 23 | VII | 8 | 9 | 10 | 25 | 12 | 13 |
| | **one** | 10 | Third | fourth | fifth | 23 | seventh | eighth | ninth | 10 | 25 | 12 | 13 |
| | **first** | tenth | Third | fourth | fifth | VI | seventh | ninth | ninth | tenth | - | - | - |
| | **Monday** | RS | Third | MC | Friday | MV | VII | - | - | - | - | - | - |
| | **Mon** | RS | HM | HM | HM | MV | MTP | - | - | - | - | - | - |
| | **January** | RS | cs | cs | May | 23 | cs | cs | Septembe | October | 25 | cs | |
| | **Jan** | CS | cs | cs | May | 23 | cs | cs | Sept | rs | 35 | cs | |
| | **A** | III | III | MC | Fifth | WV | VII | VIII | 9 | cx | Y | XII | iii |

Table 3: A table presenting top-1 predicted tokens under successor decoding from $\pi_D(1_s) + \pi_{\mathbb{N}}(i_{\mathrm{Rom}})$. Green cells denote predictions which match their target $((i+1)_s)$ exactly; yellow cells denote predictions which match the target *index* but not the target domain; red cells denote incorrect predictions. Dashed cells denote pairs of $1_s$ and $i_{Rom}$ for which $i_s$ is not a valid (single) token.

This drop in performance when switching from output-space to successor decoding (and in particular, the leakage of Roman-numeral information into $\pi_D(1_s) + \pi_{\mathbb{N}}(i_{\mathrm{Rom}})$ – notice the VII and VIII in Table 3) suggests our numeric projection $\pi_{\mathbb{N}}$ might be capturing slightly more than just the numeric subspace. Specifically, there may be some components of domain-space which are ignored by output-space decoding, but which our successor head lifts into output-space.

## P    ADDITIONAL ARITHMETIC TABLES

We display additional arithmetic tables analogously to those in Figure 6, with 3 number tables (randomly sampled ranges) in Figure 24 and a number word table in Figure 25. We see that the results are similar to those in Figure 6.

---

[8]Note, though, that, as our successor dataset contains 1041 tokens, a random classifier (even when restricted to tokens in the successor dataset) would achieve an accuracy of 0.001.

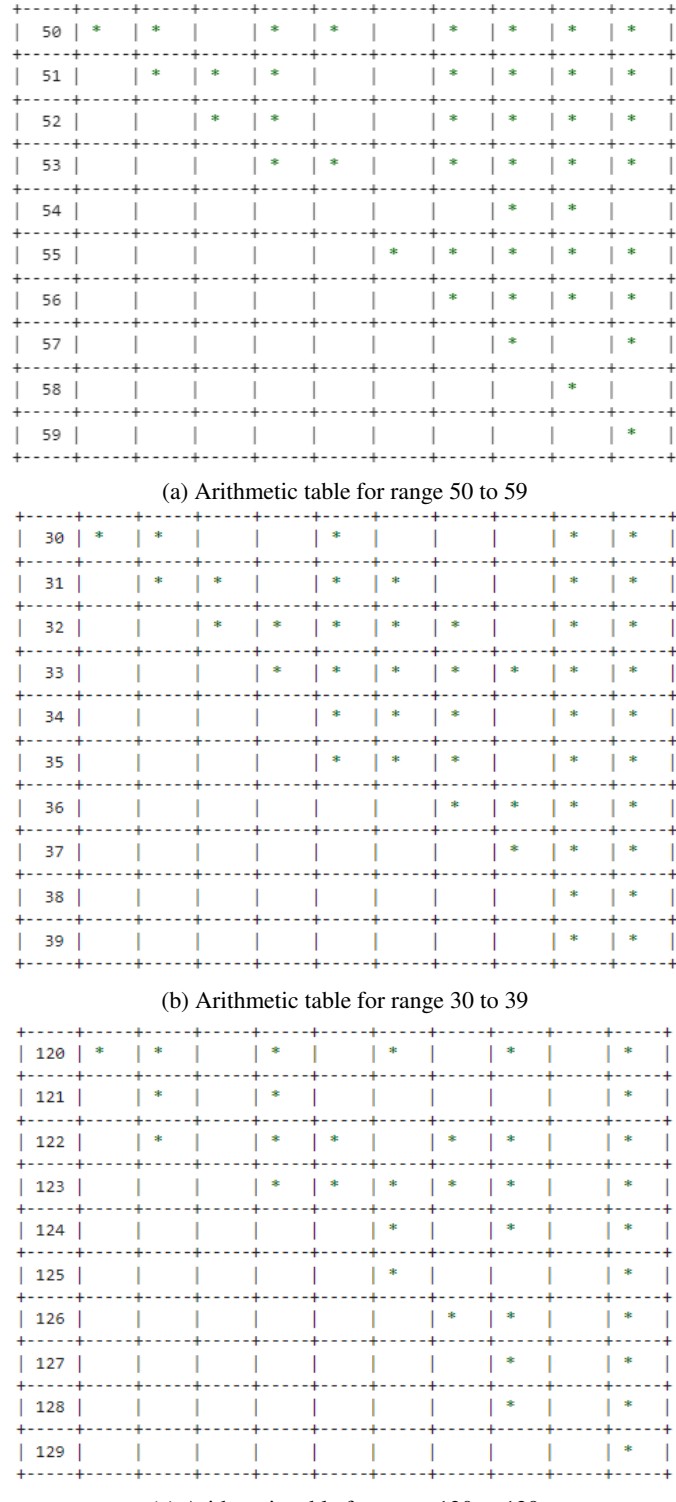

(a) Arithmetic table for range 50 to 59

(b) Arithmetic table for range 30 to 39

(c) Arithmetic table for range 120 to 129

Figure 24: Additional randomly selected number arithmetic tables, analogous to those in Figure 6

```
+-------+----+----+----+----+----+----+----+----+----+----+
|       | 0  | 1  | 2  | 3  | 4  | 5  | 6  | 7  | 8  | 9  |
+=======+====+====+====+====+====+====+====+====+====+====+
| zero  |    |    | *  | *  | *  |    | *  | *  |    | *  |
+-------+----+----+----+----+----+----+----+----+----+----+
| one   |    | *  | *  | *  | *  |    |    |    |    |    |
+-------+----+----+----+----+----+----+----+----+----+----+
| two   |    |    | *  | *  | *  | *  |    |    |    |    |
+-------+----+----+----+----+----+----+----+----+----+----+
| three |    |    | *  | *  | *  | *  | *  | *  |    |    |
+-------+----+----+----+----+----+----+----+----+----+----+
| four  |    |    |    |    | *  | *  | *  | *  | *  |    |
+-------+----+----+----+----+----+----+----+----+----+----+
| five  |    |    | *  |    | *  | *  | *  | *  | *  | *  |
+-------+----+----+----+----+----+----+----+----+----+----+
| six   |    |    |    |    |    | *  | *  | *  | *  | *  |
+-------+----+----+----+----+----+----+----+----+----+----+
| seven |    |    |    |    |    |    | *  | *  | *  | *  |
+-------+----+----+----+----+----+----+----+----+----+----+
| eight |    |    |    |    |    |    |    | *  | *  | *  |
+-------+----+----+----+----+----+----+----+----+----+----+
| nine  |    |    | *  |    | *  |    |    |    | *  | *  |
+-------+----+----+----+----+----+----+----+----+----+----+
```

(a) Arithmetic table for range zero to nine.

Figure 25: An additional arithmetic table for number words, analogous to those in Figure 6

## Q  LOGIT DISTRIBUTION FOR CELLS IN THE ARITHMETIC TABLE

The logit distributions across tokens for randomly sampled correct arithmetic examples are displayed in Figure 26.

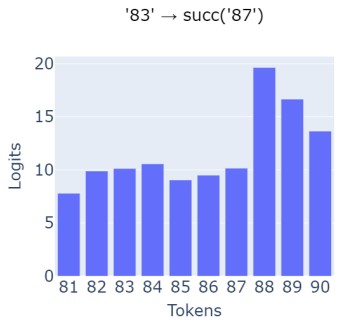

(a) Source token '83', target residue of 7 modulo 10.

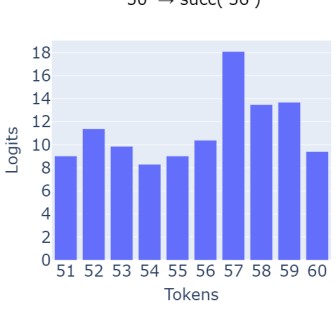

(b) Source token '50', target residue of 6 modulo 10.

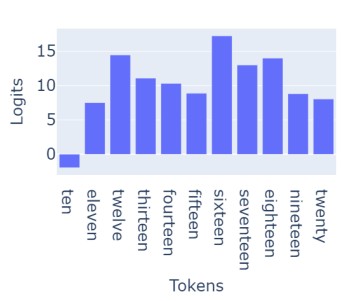

(c) Source token ' eleven', target residue of 5 modulo 10.

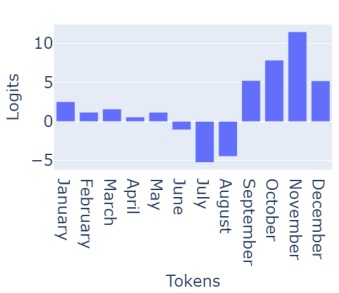

(d) Source token 'September', target residue of 0 modulo 10.

Figure 26: Logit distributions for randomly sampled checkmarked cells, sampling two cells for numbers, one cell for number words, and one cell for months.

# R    LINEAR PROBES FOR MODULI OTHER THAN $10$

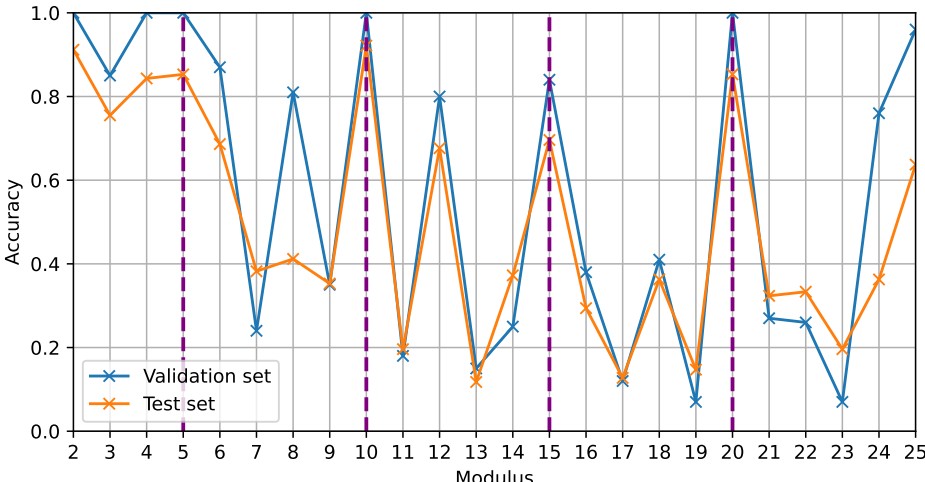

Figure 27: Validation (in-distribution task) and test (out-of-distribution task) performance for linear probes $P^{(m)}$. The vertical purple lines denote moduli divisible by 5.

While our experiments in Section 3 focus on the study of mod-10 features, one might hypothesise that there exist similar natural mod-$k$ features for other $k$. To explore this hypothesis, we repeat the linear probing experiment detailed in Section 3.2 for a range of moduli $m \in \{2, ..., 25\}$. Specifically, we test whether we can learn linear probes $P^{(m)} : \mathbb{R}^{m \times d_{\text{model}}}$ to predict the value of $i \bmod m$ from the MLP-representations of numeric tokens $t_i$, and whether these probes generalise to non-numeric tokens. As per Appendix C.1, we train our probes on numeric tokens from '0' to '500' (both with and without a leading space), holding out 10% of these tokens as a validation set, and we test our probes on a dataset of unseen tasks including cardinal numbers, Roman numerals, months and days.

We present the results of these experiments in Figure 27. Observe that, while for almost all moduli, validation and test performance are above random chance, we cannot easily extract 'mod-$m$ data' from token representations for all $m$. Indeed, the only probes with out-of-distribution performance above 0.5 are those for moduli 2-5, 6, 10, 12, 15 and 20 (and the six probes with the lowest validation performance are those corresponding to the six primes between 5 and 25).

We see some evidence, however, that *10 is a particularly significant modulus for token MLP$_0$-representations*: indeed, $P^{(10)}$ has both the joint highest validation performance (together with $P^{(2)}$, $P^{(4)}$, $P^{(5)}$ and $P^{(20)}$, and the highest out-of-distribution performance. Moreover, of all the probes whose validation accuracy is above 0.5, $P^{(10)}$ has the smallest drop in performance from in-distribution to out-of-distribution tasks.

In particular, for weeks (which we might expect to have 'mod-7 features') and months (which we might expect to have 'mod-12 features'), not only are the performances of $P^{(7)}$ and $P^{(12)}$ substantially lower than that of $P^{(10)}$, but $P^{(7)}$ fails to correctly identify the index $\bmod 7$ of any day of the week, while $P^{(12)}$ only correctly identifies the index $\bmod 12$ of 8/12 months (failing on February, May, July and August). By contrast, $P^{(10)}$ correctly identifies the index $\bmod 10$ of 4/7 days of the week, and 10/12 months.

