# OpenReview forum: "Successor Heads: Recurring, Interpretable Attention Heads In The Wild"
_ICLR.cc/2024/Conference — ICLR 2024 poster_

### Official Review · Reviewer_uBkt · 2023-10-31

**Soundness:** 4 excellent
**Presentation:** 2 fair
**Contribution:** 3 good
**Rating:** 8
**Confidence:** 3

**Summary:**

The paper leverages the transformers circuits work (Elhage '21) to uncover successor heads, which are attention heads in a transformer responsible for incrementing words / numbers which have a natural ordering. To do so, the authors build an Output-Value circuit for these successor heads, which unlike (Elhage '21), leverages a non-linear transformation of the word embeddings. The authors show that

1. successor heads occur in many decoder-only transformer architectures, and across different model scales, and different types of incrementations
2. Through several analyses and ablations (the use of linear probing on the MLP0 features, as well as the use of sparse autoencoder to extract important features for successor heads), the authors argue that the model's internal representaiton for numerical / incremental tasks is in modulo 10, i.e. that successor heads, when given input $f_{i}$, will increase the logits of $f_{i+1 \  \text{mod} \  10}$.
3. The authors show that these successor representations are amenable to vector arithmetic operations, and that while the MLP$_0$ representations are not biased towards incrementation, the OV circuit for successor head is.
4. The authors argue that successor heads enable the identification of polysemantic behavior, finding evidence that they are responsible for both acronym prediction and incrementation.

**Strengths:**

1. The empirical rigor of this work is high. The authors provide several ablations to argue the existence of mod_$10$ features in transformers. They moreover provide detailed additional information for experiments in the appendix.
2. The finding of successor heads is interesting and provides a good framework for understanding how transformers reason about incrementation.
3. The connection between incrementation and acronym prediction observed in successor heads is interesting.

**Weaknesses:**

1. The paper lacks a proper background section. Terms like OV matrix are not introduced, and more generally, the notion of circuits, or what un enembedding is, are never properly defined. This makes the paper very hard to digest without being familiar with the concepts of transformer circuits (Elhage '21). Given that the paper is already quite dense, you can e.g. move figure 7 to the appendix, and properly lay the appropriate terminology to understand this work.

**Questions:**

1. Effective OV circuit : what is meant by effective here ?
2. What was the motivation for using MLP$_0$ rather than the original word embeddings ? Given that MLP$_0$ occurs after a first self-attention layer, I would have not expected it to map embeddings to a representation amenable for the analysis presented in the paper.

On section 4 :
3. successor heads in the wild. I am not sure I understand why the authors distinguish between features that, when ablated, make the correct prediction less likely, and features that, when ablated, increase loss the most. I understand that these two are not exactly the same thing, but I don't understand what we gain by treating them differently.
4. It is mentioned that 128 samples are sampled from the Pile dataset. Did you happen to find a token where the successor head was important for each of the 128 samples ? Or did you bias your sampling towards extracts where the successor head is important ?

---

> ### Author Response · Authors · 2023-11-17
> **Rebuttal to reviewer uBkt**
>
> Thank you for the review and close understanding of our work. We hope we can address all of your comments.
>
> > The paper lacks a proper background section.
>
> As addressed in our general comment, we have added a “notation” paragraph and define terms before they are used.
>
> > Effective OV circuit : what is meant by effective here ?
>
> We use “effective” to emphasise that i) this includes MLP0, which is not standard in prior work and ii) this more closely matched how the model uses attention heads to transform information from embeddings to unembeddings
>
> > What was the motivation for using MLP rather than the original word embeddings ?
>
> Lots of literature on Language Models has found that MLP0 is crucial for model representations, e.g Wang et al., 2022 (https://arxiv.org/abs/2211.00593), Hanna et al., 2023 (https://arxiv.org/abs/2305.00586). Further, Pythia models use parallel attention (introduced in https://arxiv.org/pdf/2204.06745.pdf), ie the input to MLP0 is just the embedding and does not include the attention layer so it is even more natural for this architecture to not just use the word embeddings.
>
> > I am not sure I understand why the authors distinguish between features that, when ablated, make the correct prediction less likely, and features that, when ablated, increase loss the most.’:
>
> Winning cases tell us about the tasks for which the successor head is the primary contributor compared to other heads. In contrast, describing loss reducing cases of the successor head is independent of how other heads behave on a task. Though we believe including an analysis of winning cases is important to describe the dominating behaviour of the successor head, a description of behaviour outside of such extreme cases is also necessary to give a full picture of behaviour, and this is the goal of our loss reducing experiments.
>
> > Did you happen to find a token where the successor head was important for each of the 128 samples ? Or did you bias your sampling towards extracts where the successor head is important ?’:
>
> We apologize for not being clearer here; the 128 samples from the Pile were randomly sampled (with no bias), and we have now specified this.
>
> Is this is a satisfactory reply to your concerns?

---

> ### Author Response · Authors · 2023-11-20
> **Gentle reminder**
>
> Hello reviewer uBkt - as the end of the discussion is  **in two days time on November 22** we would like to gently remind you that **we've posted our response to your review** and hope you'll let us know if we have addressed your concerns. If you have any remaining questions, we would love to continue the discussion.

---

> > ### Comment · Reviewer_uBkt · 2023-11-20
> > **Re: Discussion**
> >
> > I am happy with the clarifications provided by the authors, and the changes made to the manuscript. I will update my score as such.

---

### Official Review · Reviewer_ExhM · 2023-10-31

**Soundness:** 3 good
**Presentation:** 3 good
**Contribution:** 1 poor
**Rating:** 3
**Confidence:** 5

**Summary:**

This paper demonstrates findings of a set of attention heads called successor heads that perform incrementation on tokens from ordinal sequences. It also shows evidence, in MLP0 layers, for "mod-10" features which are present in tokens belonging to the same numerical index. Experiments were performed to modify numeric inputs using vector arithmetic with the mod-10 features. Finally, the paper analyzes the polysemanticity of successor heads on natural language data samples. The paper's authors claim that these findings across several models of various scales demonstrate a weak form of universality.

**Strengths:**

Strengths:
1. The paper is clear and easy to read.

2. To show claims of a weak form of universality, the paper thoroughly tests for successor scores across several models for various numbers of parameters.

3. The experiments to find and verify the mod-10 features are also thoroughly performed. These features were confirmed by several different methods: first by training a sparse autoencoder using reconstruction loss on the MLP0 outputs, and then by further comparisons to linear probing and ablation to reinforce these observations. This is an interesting result uniting various tokens under mod-10 classes.

4. There are also interesting results with the natural language experiments that demonstrate interpretable polysemanticity.

**Weaknesses:**

Weaknesses:
1. In Section 4, direct effect mean ablation is used to show "that when the successor head is contributing usefully, the prompts often required some kind of incrementation". Then in Appendix J, direct effect mean ablation is used again to show that the successor head is the most important head across 64 prompts. Though this is stated with some evidence, not enough quantifiable evidence is shown here to justify the reach of these claims, such as the statement of "mostly solved". More analysis can also be shown about the "direct effect" to separate it from "indirect effects".

The paper also did not clarify the details of the ablation, such as if it used resampling ablation (there may be issues if it used mean ablation from the same dataset, as there are known issues with mean ablation [1]), and/or path patching (to obtain direct effects).

[1] https://www.lesswrong.com/posts/kcZZAsEjwrbczxN2i/causal-scrubbing-appendix#3_Further_discussion_of_zero_and_mean_ablation

2. The paper mentions that vector addition was performed successfully for 89% of the cases for digits 20-29, and mentions how it was performed on token '35'. It does not mention how this performed for other digits. This is likewise the case for number words only showing ten to twenty. Presumably, the performance is similar, but the paper should explicitly mention this to avoid criticism of cherry-picking.

In Figure 7, it's also unclear what "target residues modulo 10" means when referring to the column headers. Presumably, this is stating something similar to how the vector arithmetic on MLP0(E('fifth') makes it "behave more like MLP0(E('seventh')". The wording can be made clearer to avoid confusion that it means the number "7" rather than the word "seventh". Additionally, the checkmarks are given when "the max logits are on the successor token". This is an interesting result, but how big is the logit difference between logits for the successor token and other tokens?

Appendix D states that scaling was used on the additive feature terms. A quick explanation of why a particular scaling factor was used would be helpful.

3. The paper states: "to the best of our knowledge the presence of both successorship and acronym behavior in head L12H0 is the cleanest example of polysemantic behavior identified so far in an LLM." Why is this the cleanest example of polysemantic behavior, compared to other studies on the topic such as in [2]? Similarly for this statement, "which to the best of our knowledge are the most closely studied components in LLMs that occur in both small and large models", what other components are you comparing to that are not as closely studied?

[2] Wes Gurnee, Neel Nanda, Matthew Pauly, Katherine Harvey, Dmitrii Troitskii, and Dimitris Bertsimas. Finding neurons in a haystack: Case studies with sparse probing, 2023

4. This paper discovers novel and interesting observations, but it does not elaborate much on why this observation is impactful enough.

**Questions:**

Questions:
1. While using MLP0 with successor heads, in isolation, was shown to be sufficient to perform incrementation, it is not shown how they interact with the rest of the models they belong to that the paper studied. How is this information about numerical features processed in later layers, from MLP0 to the successor head L12H0, then to the rest of the model? How does this end-to-end path interact with alternative paths, and inhibitory ones?

The mod-10 features are obtained from MLP0's outputs, to study how the end-to-end path of MLP0 to the successor head processes these features. However, the paper does not show how this information is processed through other layers and MLPs in the models. Can such mod-10 features be found in other layers?

2. To continue on weak point #2:
There are also the cases of adding non-mod-10 features to numeric tokens, and adding the features to non-numeric tokens. There may be cases of "confounding" factors where U(OV(MLP0(E('twelve')) - f_2 + f_4)) has high logits for 'fourteen', but is it possible that adding non-numeric features will also shift twelve to thirteen, fourteen, etc.? If so, then perhaps these mod-10 features are just obtaining the correct change because of how the numeric features are scaled with one another? In other words, would subtracting non-numeric features R then adding RR obtain the same result? This is not a weak point of the paper as it may be beyond its scope, and this situation seems unlikely to be the case, but it can be further investigated for thoroughness.

3. To continue on weak point #3:
In terms of interpretable polysemanticity, how do other heads compare to successor heads? To ensure acronym handling is not commonly done for many attention heads, how well do other heads handle acronyms?


- Other comments:

This paper tackles a similar topic as a previous project from earlier this year, which also found a successor attention head by performing OV circuit analysis on inputs of numbers, number words, days, months, and letters. It also was stated to have inspected the effects of vector addition on number tokens [3]. Could the authors elaborate on the similarities and differences?

[3] https://alignmentjam.com/project/one-is-1-analyzing-activations-of-numerical-words-vs-digits

---

> ### Author Response · Authors · 2023-11-17
> **Rebuttal to reviewer ExhM (1/2)**
>
> Thank you for your kind words about our paper. We hope we can clarify all your questions, please reply if there are more issues. We draw your attention to a new Appendix (L) that does more ablation experiments in response to your concerns.
>
> > not enough quantifiable evidence is shown here to justify the reach of these claims, such as the statement of "mostly solved”
>
> This is a good point, and we agree that the indirect effects should have been studied. We have now analysed the indirect and the total effect for the loss reducing experiments, and evidence that indirect effects do not play a significant role in successorship. These results are described in Appendix L.
>
> > The paper also did not clarify the details of the ablation
>
> We apologize for not specifying this. Our experiments in Section 4 used direct effect mean ablation, taking the mean on the same batch, and we have now included this detail. Additionally, we have repeated our loss reducing experiments using resampling ablation instead of mean ablation and see similar results, with results described in Appendix L.
>
> > Presumably, the performance is similar, but the paper should explicitly mention this to avoid criticism of cherry-picking
>
> We have now mentioned that performance is similar across other tokens and have included some additional arithmetic tables, displayed in Appendix O.
>
> > it's also unclear what "target residues modulo 10" means when referring to the column headers
>
> We have included an example of what we mean by ‘target residue’ in the caption of Figure 6 (this figure was Figure 7). We hope this makes things clearer.
>
> > how big is the logit difference between logits for the successor token and other tokens?
>
> We have included the logit distributions for randomly sampled checkmarked cells in Appendix P to give a picture as to how the logits differ across tokens.
>
> > A quick explanation of why a particular scaling factor was used would be helpful
>
> We agree that this aspect should be described further and have added some more explanation behind choosing the scaling factor in Appendix D.
>
> > what other components are you comparing to that are not as closely studied?
>
> We are comparing to induction heads (https://transformer-circuits.pub/2022/in-context-learning-and-induction-heads/index.html), the only other interpretable, recurring language model component in small and large models that we claim do not have as crisp an explanation as successor heads (analyzing attention patterns is not an end-to-end explanation) - we have now reviewed this in the related work section.
>
> > Why is this the cleanest example of polysemantic behavior?
>
> Thank you for the reference to the Gurnee et al. work. We agree that this provides helpful evidence about polysemanticity in neurons. We think that our work is cleaner, since we measure the loss explained by the succession, acronym and other behaviors (Figure 10). This shows that this attention head’s direct effect has over half of its behavior explained by these behaviors – Gurnee et al show that some neurons implement several different tasks, but don’t make claims about the role of these neurons across the whole training distribution. It is possible that their bigram examples do not explain the majority of the role of this neuron in the model.
>
> > This paper discovers novel and interesting observations, but it does not elaborate much on why this observation is impactful enough
>
> We agree that the primary contribution of our work is about interesting phenomena without a directly impactful outcome. However, as elaborated in the introduction, there are several second-order effects of our work. Firstly, we provide useful evidence about the universality of representations in models: successor heads generalize to larger models, but cross-task generalization is more complicated, such as how steering the successor head on days of the week did not produce as strong results. Additionally, we find evidence that models have compositional representations, such as combining a mod 10 feature with task-based information to produce certain completions, suggesting that LLMs have sophisticated internal structure, a similar conclusion to https://arxiv.org/abs/2310.02207, but from a different angle (numeracy). Finally, our work shows another bit of evidence for the strength of Sparse Autoencoders, which in parallel to our work were found to have a lot of success at almost completely explaining a small language model (https://transformer-circuits.pub/2023/monosemantic-features), which we hope the interpretability field will find useful.
>
> (This rebuttal is continued in "Rebuttal to reviewer ExhM (2/2)")

---

> ### Author Response · Authors · 2023-11-17
> **Rebuttal to reviewer ExhM (2/2)**
>
> (Continued from "Rebuttal to reviewer ExhM (1/2)")
>
> > How is this information about numerical features processed in later layers, from MLP0 to the successor head L12H0, then to the rest of the model? How does this end-to-end path interact with alternative paths, and inhibitory ones? (...) the paper does not show how this information is processed through other layers and MLPs in the models.
>
> We believe that a full exploration of these details is out-of-scope for our work, however we have added analysis of the indirect effect to a new Appendix to our work (Appendix L). This suggests that processes occurring after the successor head’s output may not play a significant role to succession, hence an analysis beyond the direct effect is less important.
>
> > would subtracting non-numeric features R then adding RR obtain the same result?
>
> We are not sure we understand this concern. We have included additional experiments exploring the separability of the numeric and task subspaces in Appendix M, which we hope you appreciate. These demonstrate that we can isolate a common numeric subspace within embedding space, that for any given token (e.g. ‘February’), encodes the index of that token within its ordinal sequence (e.g. months). If this does not address your concern, let us know.
>
> > To ensure acronym handling is not commonly done for many attention heads, how well do other heads handle acronyms?
>
> We consider this investigation out of scope, and not part of our main contribution of our work. The results on acronyms are intended to show that 1) attention head polysemanticity can be observed in practice but that 2) it is not necessarily an intractable phenomena. Acronym behavior accounts for over 10% of the loss recovered by the successor head. Acronyms occupy less than 1% of the pretraining data and therefore it is very unlikely that this attention head is not one of the most important model components for completing acronyms. We are excited for further research into attention head polysemanticity and hope our work spurs this on.
>
> > Could the authors elaborate on the similarities and differences?
>
> Though the referenced project similarly studies model behaviour on tasks involving incrementation and finds a successor head, it is limited to a single successor head in a small model (GPT-2 Small), whereas our work finds that this is a more general and recurring phenomena in larger models of varying architecture, and we describe an end-to-end circuit that can perform incrementation using these heads. Our work also performs a weight-level analysis of successor heads, allowing us to discover mod-10 features, however the referenced project does not perform such an analysis. They also do not investigate how their found successor head performs on natural language datasets and the tasks for which it is important for, which we analyze in Section 4. We were not aware of this work (and it is outside the time period of necessary consideration and non-archival) but have added it as a reference and are glad that some of our results were found independently.
>
> Have we addressed all further concerns you had with our work?

---

> > ### Comment · Reviewer_ExhM · 2023-11-21
> > **Response to reviewers**
> >
> > Thank you for taking the time to respond to my questions and comments. I find the responses helpful, apart from the final comment on similarities with previous work, which overlaps with this work. I will maintain my score.

---

> > > ### Comment · Reviewer_ExhM · 2023-11-22
> > > **Further responses to the authors**
> > >
> > > Since the authors did not cite nor acknowledge the large overlap between this work and [1], I will adjust my score. [1] was published mid-July which is well within the period of necessary consideration.
> > >
> > > [1] - https://alignmentjam.com/project/one-is-1-analyzing-activations-of-numerical-words-vs-digits

---

> > > > ### Author Response · Authors · 2023-11-22
> > > > **Misunderstanding**
> > > >
> > > > Thank you for your continued engagement. **We think your last comment is mistaken**
> > > >
> > > > > [1] was published mid-July which is well within the period of necessary consideration
> > > >
> > > > From https://iclr.cc/Conferences/2024/ReviewerGuide
> > > >
> > > > > *if a paper was published (i.e., at a peer-reviewed venue) on or after May 28, 2023, authors are not required to compare their own work to that paper*
> > > >
> > > > **This weekend hackathon project was not peer reviewed** and was published after May 28. Further you said
> > > >
> > > > > the authors did not cite nor acknowledge the large overlap between this work and [1]
> > > >
> > > > We already updated the manuscript and cited the work; based on this disagreement, we added one further detail. In our rebuttal we also already explained in detail the similarities, and moreover the differences to this work e.g "our works finds [*successor heads*] ... in larger models of varying architecture, and we describe an end-to-end circuit".
> > > >
> > > > **Due to this misunderstanding, would you be open to changing your score back to your original assessment, that our paper was worth accepting?**

---

> ### Author Response · Authors · 2023-11-20
> **Gentle reminder**
>
> Hello reviewer ExhM - as the end of the discussion is  **in two days time on November 22** we would like to gently remind you that **we've posted our response to your review** and hope you'll let us know if we have addressed your concerns. If you have any remaining questions, we would love to continue the discussion.

---

> ### Comment · Reviewer_ExhM · 2023-11-22
> **Respond to misunderstanding**
>
> While the citation issue is noteworthy, focusing solely on publication status fails to account for influential unpublished work, which plays a vital role in scientific progress. Basing judgments of novelty on publication date alone seems questionable when substantive overlap exists between ideas and findings. Though understandable given the recency of some relevant cautionary findings, the argument that a particular study entered the public domain after a given date does not convincingly address fundamental concerns about the degree of new insights offered here.
>
> I still believe this is solid research, but in reassessing its novelty through comparison with previously proposed work, I find that key contributions appear modest. Consequently, I have reconsidered the scores assigned, aiming to better reflect the incremental nature of what has been achieved. I hope this explanation helps clarify my perspective.

---

### Official Review · Reviewer_Vpua · 2023-11-01

**Soundness:** 3 good
**Presentation:** 4 excellent
**Contribution:** 3 good
**Rating:** 8
**Confidence:** 4

**Summary:**

This paper discovers and analyzes Successor Heads, a type of attention heads in transformer language models that increases the probability of the next token in a sequence such as 1 -> 2, or Monday -> Tuesday. This is discovered mainly through ablations on a specific dataset crafted for this task, and analyzed through various means.

**Strengths:**

Good informative figures such as Fig 1 and Fig 7, clear writing. The use of OV circuits in the discovery and analysis seems smart and somewhat novel to standard methodology for these kinds of findings. Interesting behavior and good multi-pronged analysis of it.

**Weaknesses:**

Somewhat overclaiming the contribution:
For example abstract says:  "Existing research in this area has found interpretable language model components in small toy models. However, results in toy models have not yet led to insights that explain the internals of frontier models and little is currently understood about the internal operations of large language models." This makes it sound like existing work has only studied toy models which is not true, while also making it sound like this work would study frontier models which is not the case. While they look at larger models than most related work, the wording makes it sound like difference is larger than it is.

Also the findings about mod 10 features are almost entirely based on the setting of incremental numbers which makes sense, while the writing makes it sound like they are behind successor head behavior on all tasks. The only evidence of these being used on other task is a low success percentage on changing output month with vector arithmetic. I would expect for tasks like months and days there would be other mod-12 or mod-7 features for example that could explain this behavior, was this studied?

**Questions:**

How was the set of succession datasets chosen? Did you experiment with other tasks that were eventually not included? It would be interesting to measure successor behavior on some held out succession task, as currently behavior on all the tasks was used to find successor heads.

---

> ### Author Response · Authors · 2023-11-17
> **Rebuttal to reviewer Vpua**
>
> We appreciate your engagement with our work and hope we can address your concerns.
>
> > This makes it sound like existing work has … which is not true, while also making it sound like this work … which is not the case
>
> Thank you for your feedback. We agree that we were not clear about how our work improves on existing work. We intended to emphasise how a mechanistic understanding of anything in a large, non-toy model is unusual. We have slightly rephrased this (within ICLR 2024 policy) to say
>
> *Existing research in this area has struggled to find recurring, mechanistically interpretable language model components beyond small toy models. Further, existing results have led to very little insight to explain the internals of larger models that are used in practice.*
>
> and we have also added to the “Transformer Circuits” part of the related work.
>
> > The only evidence of these [mod 10 features] being used on other task is a low success percentage on changing output month
>
> We disagree with this statement. In Section 3.1 we show our results on Linear Probing, where we find that probing for mod 10 features generalizes to test data, and even to unseen tasks, with high accuracy. Our paper was also upfront about the limitations of steering with mod 10 features at the end of Section 3, but the mod 10 results are stronger (see also the next answer).
>
> > I would expect for tasks like months and days there would be other mod-12 or mod-7 features
>
> To investigate this, we trained linear probes (as per Section 3.2) for all moduli from 2 to 25. Of all probes tested, we found that the mod-10 probe had the **joint highest in-distribution accuracy** (together with the mod-2, 4, 5 and 20 probes) and the **highest out-of-distribution accuracy**: not only is ‘mod-10 data’ easiest to extract from MLP0-representations, but the relevant features generalise well.
>
> We also found that the mod-7 and mod-12 probes **performed worse than the mod-10 probe** at predicting the residues of weeks and months respectively.
>
> Full details and results for this experiment are provided in Appendix Q.
>
> > How was the set of succession datasets chosen?
>
> We included all single tokens we could find that were involved in listing, though kept Roman numerals as a held out task.
>
> > Did you experiment with other tasks that were eventually not included?
>
> We used Roman numerals as a held out test case for Linear Probing, and found that the model still performed well. We extended our study of this task to evaluate succession score in response to your comment, and show these further strong results in Appendix B2.

---

> > ### Comment · Reviewer_Vpua · 2023-11-22
> > **Response to rebuttal**
> >
> > Thanks, this mostly addresses my concerns.
> >
> > > In Section 3.1 we show our results on Linear Probing, where we find that probing for mod 10 features generalizes to test data, and even to unseen tasks, with high accuracy
> >
> > I still think this only shows that mod 10 features are recoverable from the presentation, but it does not show that the features are important to the successor head on these other tasks. However I agree upon rereading section 3 it is relatively upfront on the limitations in the end.
> >
> > I will increase my score accordingly.

---

> ### Author Response · Authors · 2023-11-20
> **Gentle reminder**
>
> Hello reviewer Vpua - as the end of the discussion is  **in two days time on November 22** we would like to gently remind you that **we've posted our response to your review** and hope you'll let us know if we have addressed your concerns. If you have any remaining questions, we would love to continue the discussion.

---

### Official Review · Reviewer_uo7e · 2023-11-04

**Soundness:** 3 good
**Presentation:** 2 fair
**Contribution:** 3 good
**Rating:** 8
**Confidence:** 3

**Summary:**

This paper digs inside the mechanism of attention heads in LLMs and discovers some particular attention heads are able to fire for predicting naturally-ordered tokens, which is termed as successor heads. This paper belongs to a recent line of work, the mechanical interpretability of transformer models. The findings of successor heads appear to be common for different prompts and also across models, showing some level of polysemanticity in the successor attention heads’ activation space.

**Strengths:**

The findings presented in this paper are significantly novel. Authors have clearly described the functions of successor heads and designed multiple experiments to validate their hypothesis. I especially appreciate Section 3.3 where the evidence in arithmetic is a strong proof that the activation of success attention indeed captures the natural ordering of words and is responsible for the LLM’s reasoning.

**Weaknesses:**

There are a few issues mostly in the presentation of the work.

I am getting really annoyed when the authors place all definitions, i.e. the Glossary section, at the end of the appendix. It is really inconvenient for the reader to go back and forth during the reading. It has to have better ways to present the definitions in the context. Please do not do this.

It lacks sufficient descriptions for the reader to understand the process that parses the original output of the attending heads to the sparse encoder’s output. I understand that this is to make more room to present the findings; however, it makes the methodology part pretty unclear from reading the current version. I have to go back and forth and spend a lot more time on Section 2 and 3 to make sure I understand the way each figure is plotted.

**Questions:**

How does the choice of n in top-n-attended tokens affect the findings? Authors only pick a particular k (i.e. k=5 or k=1) and present the result. Can you demonstrate the full story about polysemantics when you relax the constraint of importance by choosing large n?

---

> ### Author Response · Authors · 2023-11-17
> **Rebuttal to reviewer uo7e**
>
> We would like to thank the reviewer for their thoughtful comments and positive assessment of our work! We respond accordingly to your comments and questions below.
>
> > It is really inconvenient for the reader to go back and forth during the reading.:
>
> As described in our global response, we have addressed your concern by defining terms the first time we use them, and moving the glossary to before the appendices. We hope this change makes reading the paper more convenient.
>
> > It lacks sufficient descriptions for the reader to understand the process that parses the original output of the attending heads to the sparse encoder’s output.:
>
> We have modified our description of the sparse encoder experiments to further clarify the methodology of Section 3. Additionally, we have further elaborated on our use of SAEs in Appendix A.2, including a diagram to illustrate the process.
>
> > Can you demonstrate the full story about polysemantics when you relax the constraint of importance by choosing large n?’:
>
> This is an interesting question. We ran the loss reducing experiments (described by Figure 10) but instead using top-50 for each behaviour and this gave the loss proportions: 39.6% for successorship, 20.4% for acronym, 12.3% for greater-than, and 20.8% for copying behaviour, which accounts for a total of 93.1% of behaviour, hence using a large $n$ does appear to capture most behaviour. A slightly stronger condition of top-25 still accounts for a total of 85.8% of behaviour.
>
> Please ask at any time if you think we could improve further.

---

> ### Author Response · Authors · 2023-11-20
> **Gentle reminder**
>
> Hello reviewer uo7e - as the end of the discussion is  **in two days time on November 22** we would like to gently remind you that **we've posted our response to your review** and hope you'll let us know if we have addressed your concerns. If you have any remaining questions, we would love to continue the discussion.

---

### Author Response · Authors · 2023-11-17
**Rebuttal - general comment**

We thank the reviewers for all recommending acceptance, and for the comments that our work was “significantly novel” (reviewer uo7e) through a “multi-pronged analysis” (reviewer Vpua).

We apologise for some of our notation and structure being confusing, which many reviewers were concerned by.

To rectify this,
1. We have ensured that we define all terms the first time we use them.
2. We have moved the glossary before the Appendix so it is easier to read. Moreover, given the changes made by point (1), we expect readers to need to use the glossary less.

We use red pen to clearly show the changes based on feedback in our reuploaded PDF.

The reviewers had many interesting questions about our experiments, and we were very excited to address almost all of these in extensive comments and appendices. We hope that this provides even more evidence for the generality and interpretability of successor heads.

---

### Meta-Review · Area_Chair_BzvF · 2023-12-10

**Metareview:**

This paper presents empirical findings of the attention heads (termed successor head) in LLMs based on extensive experiments. All the reviewers agree the finding is interesting and give insights to understand LLMs better through mechanical interpretability. There was a concern on the recent hackathon work that has similar findings on the successor head, and the authors have acknowledged the work and cite in the revised draft with a short discussion. It is suggested that the authors to expand the discussion on the difference and similarities shared by this contemporary work for better clarity in the camera ready version.

The recommendation for this paper is accept.

**Justification For Why Not Higher Score:**

while interesting, the finding in this paper is purely empirical and has limited scope.

**Justification For Why Not Lower Score:**

this paper provides some interesting insights from empirical experiments.

---

### Decision · Program_Chairs · 2024-01-16

Accept (poster)